



# Organic aerosol source apportionment in Zurich using an extractive electrospray ionization time-of-flight mass spectrometry (EESI-TOF): Part II, biomass burning influences in winter

Lu Qi[1,2], Mindong Chen[2], Giulia Stefenelli[1], Veronika Pospisilova[1], Yandong Tong[1], Amelie Bertrand[1],
5   Christoph Hueglin[3], Xinlei Ge[2], Urs Baltensperger[1], André S. H. Prévôt[1], Jay G. Slowik[1]

[1]Laboratory of Atmospheric Chemistry, Paul Scherrer Institute (PSI), Switzerland
[2]Nanjing University of Information Science & Technology, 21000 Nanjing, China
[3]EMPA, Dübendorf 8600, Switzerland

*Correspondence to:* Jay G. Slowik (jay.slowik@psi.ch) and André S. H. Prévôt (andre.prevot@psi.ch)


**Abstract.**

Real-time, in situ molecular composition measurements of the organic fraction of fine particulate matter (PM$_{2.5}$) remain challenging, hindering a full understanding of the climate impacts and health effects of PM$_{2.5}$. In particular, the thermal decomposition and ionization-induced fragmentation affecting current techniques has limited a detailed investigation of

secondary organic aerosol (SOA), which typically dominates OA. Here we deploy a novel extractive electrospray ionization time-of-flight mass spectrometer (EESI-TOF-MS) during winter 2017 in downtown Zurich, Switzerland, which overcomes these limitations, together with an Aerodyne high resolution time of flight aerosol mass spectrometer (HR-TOF-AMS) and supporting instrumentation. Positive matrix factorization (PMF) implemented within the Multilinear Engine (ME-2) was applied to the EESI-TOF data to quantify the primary and secondary contributions to OA. An 11-factor solution was selected

as the best representation of the data, including 5 primary and 6 secondary factors. Primary factors showed influence from cooking, cigarette smoke, biomass burning (2 factors) and a special local unknown event occurred only during two nights. Secondary factors were affected by biomass burning (3 factors, distinguished by temperature and/or wind direction), organonitrates, monoterpene oxidation, and undetermined regional processing, in particular the contributions of wood combustion. While the AMS attributed slightly over half the OA mass to SOA but did not identify its source, the EESI-TOF

showed that most (> 70 %) of the SOA derived from biomass burning. Together with significant contributions from less aged biomass burning factors identified by both AMS and EESI-TOF, this firmly establishes biomass burning as the single most important contributor to OA mass at this site during winter. High correlation was obtained between EESI-TOF and AMS PMF factors where specific analogues existed, as well as between total signal and POA/SOA apportionment. This suggests the EESI-TOF apportionment can be approximately taken at face value, despite ion-by-ion differences in relative sensitivity. The

apportionment of specific ions measured by the EESI-TOF (e.g. levoglucosan, nitrocatechol, and selected organic acids), and utilize a cluster analysis-based approach to identify key marker ions for the EESI-TOF factors are investigated. The interpretability of the EESI-TOF results and improved source separation relative to the AMS within this pilot campaign validate the EESI-TOF as a promising approach to source apportionment and atmospheric composition research.

## 1 Introduction

Organic aerosol (OA) is relevant due to its roles in several atmospheric processes including radiative forcing, visibility, heterogeneous reactions, and uncertain effects on human health (Nel, 2005; Docherty et al., 2008; Stocker, 2013). OA sources are typically classified as either directly emitted primary organic aerosol (POA) or secondary organic aerosol (SOA) formed from gas-to-particle conversion after chemical reactions. SOA is estimated to comprise approximately 20 % to 90 % of OA, depending on location and time of year (Jimenez et al., 2009; Hallquist et al., 2009). Many studies have successfully linked

POA to specific sources, but the level of chemical characterization achieved by conventional online instrumentation has been in most cases proven insufficient for quantitative resolution of SOA source contributions and/or formation pathways.



Therefore, the effects of individual SOA sources on health and climate remain poorly constrained, hampering the design of efficient emissions control policies.

A range of methods to measure molecular composition of aerosol particles have so far mostly been conducted offline, using filter samples (Wang et al., 2009; Daellenbach et al., 2017; Wang et al., 2017). Compared to online methods, offline methods have low time resolution typically integrating aerosol over hours and introducing sampling/storage artifacts (Timkovsky et al., 2015). Moreover, offline measurement techniques like gas chromatography-mass spectrometry (GC-MS) or liquid chromatography-mass spectrometry (LC-MS), are chemically highly specific, but often struggle with the fraction of mass that can be characterized (typically < 20 % of the total OA), which hinders our understanding of the SOA.

Currently available online speciation techniques to measure aerosol particle composition often rely on some type of thermal desorption and/or hard ionization leading to thermal decomposition and/or ionization-induced fragmentation of the original molecules. For example, the Aerodyne aerosol mass spectrometer (AMS) vaporizes molecules at 600 C followed by electron ionization at 70 eV, facilitating quantification but yielding extensive decomposition and fragmentation (Jayne et al., 2000; Sasaki et al., 2001; Samy et al., 2011; Hayes et al., 2013). The chemical analysis of aerosol online-proton transfer reaction mass spectrometer (CHARON-PTR-MS) has no significant thermal decomposition but the ionization scheme fragments of typical SOA molecules (Eichler et al., 2015; Muller et al., 2017). Several semi-continuous methods have also been developed, including the thermal desorption aerosol-GC (TAG-MS, family, Williams et al., 2006) and gas and aerosols-chemical ionization time-of-flight mass spectrometer (FIGAERO-CIMS, Lopez-Hilfiker et al., 2014). However, these systems remain subject to some degree of thermal decomposition, as well as potential reaction on the collection substrate, and significantly lower time resolution. Above all, an online instrument able to detect the original OA and resolve its chemical composition at the molecular level with higher time resolution is needed. The Paul Scherrer Institute (PSI) has developed such an instrument, i.e., the extractive electrospray ionization time-of-flight mass spectrometer (EESI-TOF), measuring particles at molecule level with a time resolution of seconds while overcoming the usual limitations, e.g. thermal decomposition, ionization-induced fragmentation, semi-continuous operation (Lopez-Hilfiker et al., in prep.).

Due to the lacking ability to apportion SOA to specific sources, a terminology based on properties rather than sources was previously introduced, such as the AMS-based discrimination into semi-volatile and low-volatility oxygenated organic aerosol (SV-OOA and LV-OOA, respectively). The current state-of-the art SOA source apportionment is to be improved based on large laboratory experiments which generate a "library" of species of the SOA products (Zhang et al., 2015; Bianchi et al., 2017; Nakao et al., 2011; Nah et al., 2016; Zhang et al., 2017a). An isoprene-OA source was identified based on fragments in AMS and ACSM (Aerosol Chemical Speciation Monitor) mass spectra that are consistent with those of laboratory-generated isoprene SOA (via reactive uptake of epoxydiols (IEPOX), Xu et al., 2015; Zhang et al., 2017b). Offline analysis identified winter OOA and summer OOA which to some extent appears to be linked to sources (Daellenbach et al., 2017; Daellenbach et al., 2016; Bozzetti et al., 2016), even though the corresponding sources cannot be retrieved (Lu et al, in prep.). Zhang et al., (2018) combined the offline GC-MS method and online FIGAERO-CIMS measurements to better characterize summertime monoterpene SOA.





Domestic wood combustion has been identified as a major source of OA in central Europe (Lanz et al., 2010; Herich et al., 2014), as well as in Asia (Sun et al., 2013; Quan et al., 2014). Recent studies have been devoted to the chemical characterization of the gas and particle-phase emissions from biomass burning in the laboratory, to provide information for a better source apportionment of primary and secondary biomass burning OA (Iinuma et al., 2010; Nakao et al., 2011; Ofner et

al., 2011; Chan et al., 2005; Bruns et al., 2017; Bertrand et al., 2018). Various tracer compounds for biomass burning were reported, including levoglucosan, which is a sugar anhydride compound produced from the pyrolysis of cellulose and hemicellulose (Fine et al., 2001), or methoxyphenols (e.g. guaiacol and syringol), derived from the pyrolysis of lignin (Coeur-Tourneur et al., 2009; Veres et al., 2010; He et al., 2018), and methyl-nitrocatechols, nitrated aromatic compounds from biomass burning (Iimuma et al., 2010a). Furthermore, biomass burning has been shown to produce significant SOA in

laboratory measurements (Bruns et al., 2016; Nakao et al., 2011; Yee et al., 2013; Stefenelli et al., 2019), but this component has not yet been resolved in the field with the partial exception of winter OOA.

Here, we report on a study in Zurich, a mid-size city in Central Europe, utilizing the EESI-TOF, complemented with AMS source apportionment results for a winter case. Summer measurement and source apportionment are presented in the companion paper (Stefenelli et al., in prep.). In both cases, due to the enhanced chemical resolution of the EESI-TOF we are

able to resolve more POA and SOA sources than in previous studies at the same site.

## 2 Methodologies

### 2.1 Measurement Campaign

Measurements were performed from 25 January to 5 February, 2017 at the Swiss National Air Pollution Monitoring Network (NABEL) station at Zurich Kaserne, Switzerland (Richard et al., 2011). The station is located in the center of the metropolitan

area of Zurich (1.3 million inhabitants). It is characterized as an urban background site, although several restaurants are nearby (Lanz et al., 2007). Long-term measurements at the site include ambient meteorological data such as temperature, relative humidity (RH), solar radiation, wind speed and direction, trace gas measurements comprising nitrogen oxides (NO$_x$, Thermo Environmental Instruments 42i, Thermo Electro Crop., Waltham, MA) and ozone (O$_3$, Thermo Environmental Instruments 49C, Thermo Electro Crop., Waltham, MA), and particle measurements which include size distributions (scanning mobility

particle sizer, SMPS, TSI) and number concentration (condensation particle counter, CPC).

For the intensive campaign, an EESI-TOF, an HR-TOF-AMS (Aerodyne Research Inc.) and an SMPS were additionally deployed. The sampling was performed in a mobile trailer installed outside the NABEL station. Ambient air was sampled through a PM$_{2.5}$ cyclone to remove coarse particles (~75 cm above the trailer roof and ~5 m above ground). The air passed through a stainless steel (~ 6 mm) tube into the AMS, EESI-TOF, and SMPS, installed on the same line and in close proximity.



## 2.2 Instrumentation

### 2.2.1 Extractive Electrospray Ionization Time-of-flight Mass Spectrometer (EESI-TOF)

The extractive electrospray ionization time-of-flight mass spectrometer (EESI-TOF) is a novel instrument for real-time measurement of organic aerosol without thermal decomposition or ionization-induced fragmentation. The instrument is

discussed in detail elsewhere (Lopez-Hilfiker et al., in prep.) and a brief overview is presented here. Ambient aerosol is continuously sampled at 900 cm$^3$ min$^{-1}$, either directly or through a particle filter to yield a background measurement. In this study, 10 mins ambient air sampling was alternated with 2 mins through the filter with spectra recorded with 40 s time resolution. The flow then passes through a 5 cm long 6 mm outer diameter (OD) multi-channel extruded carbon denuder housed in a stainless steel tube, which removes most trace gas phase species. The denuder eliminates artifacts from

semi-volatile species desorbing from the filter, and also improves detection limits by reducing the gas-phase background. The particle-laden flow then intersects a spray of charged droplets generated by a conventional electrospray capillary. Particles collide with the electrospray droplets and the soluble components are extracted, ionized by Coulomb explosion of the charged droplets, and detected by TOF-MS (resolution ~4000 at mass to charge ratio (*m/z*) 185). The electrospray droplets are generated by a commercially available 360 μm OD untreated fused silica capillary with an inner diameter of 50 μm (BGB

Analytic). The sample flow remains unheated until after extraction of aerosol material into the electrospray droplets, minimizing volatilization of labile particle phase components and thermal decomposition. The electrospray working solution is a 50/50 water/acetonitrile (> 99.9 %, Sigma-Aldrich) mixture, which has less background signal compared to the water/methanol mixture, with 100 ppm of sodium iodide (NaI) as a charge carrier. Spectra are recorded in positive ion mode, in nearly all cases as adducts with Na$^+$. The recorded signals are linear with mass and free of detectable matrix effects, in part due

to the suppression of ionization pathways other than Na$^+$ adduct formation (Lopez-Hilfiker et al., in prep.). Here we report the signal measured by the EESI-TOF in terms of the mass flux of ions to the microchannel plate detector (attograms s$^{-1}$, neglecting Na$^+$), calculated as shown in Eq. 1.

$$M_x = I_x \cdot (MW_x - MW_{cc}) \tag{1}$$

Here $M_x$ is the mass flux of ions united in ag s$^{-1}$, $x$ represents the measured molecular composition. $I_x$ is the recorded signal

measured by EESI-TOF. $MW_x$ and $MW_{cc}$ represent the molecular weight of the ion and the charge carrier (e.g. H$^+$, Na$^+$), respectively. Note that this measured mass flux can be related to ambient concentration by the instrument flow rate, EESI extraction/ionization efficiency, declustering probability, and ion transmission, where several of these parameters are ion-dependent.

With the EESI-TOF, we almost continuously collected data from 25 January to 5 February, 2017 (84.6 %), missing a few time

points due to instrumental calibration and issues such as cleaning the electrospray capillary due to lost or unstable signal. The determination of the sensitivity and check of the linearity with mass was achieved by calibration, directly nebulizing levoglucosan into the EESI with simultaneous quantification by an SMPS.



Data processing was executed using Tofware version 2.5.7 (Tofwerk AG, Thun, Switzerland). The total number fitting of 1125 ions (including 882 Na$^+$ adducts, one H$^+$ adduct, and 242 unknown ions) between *m/z* 135 and 400 were identified. Negligible signal was detected below *m/z* 135 due to the selected mass spectrometer transmission window. Data were pre-averaged to 1 min time resolution, and high resolution peak fitting was performed. Individual 1-min spectra were classified as either ambient

measurements, background sampling (through the particle filter), or imperfect mid-spectrum when a switch happened between servo and MS, in the latter case, points were excluded from further analysis. Background spectra were averaged across each 2-min filter period, and these filter periods were interpolated to estimate the background spectrum during each ambient sampling period. The estimated backgrounds were subtracted from individual ambient spectra to yield the final ion time series of difference spectra. Ions with a mean signal-to-noise ratio (SNR) below 2 were removed from the further analysis. No

corrections for the relative sensitivity of individual ions or drift in instrument sensitivity were applied. For the Multilinear engine (ME-2) source apportionment analysis (Sect. 2.3), data were re-averaged to 2 mins. The corresponding error matrix $\sigma_{ij}$, which has the same dimensions as the data matrix, follows the model of Allan et al. (2003), which calculation includes the uncertainty deriving from electronic noise, ion-to-ion variability at the detector and ion counting statistics. The error estimates in this case incorporate the uncertainties related to both the ambient measurements ($\delta_{ij}$) and the background ($\beta_{ij}$), which are

combined in quadrature according to Eq. 2:

$$\sigma_{ij} = \sqrt{{\delta_{ij}}^2 + {\beta_{ij}}^2} \qquad (2)$$

The final data matrix and error matrix has the size of 10165 (time series) × 892 (variables).

**2.2.2 Aerosol Mass spectrometer (AMS)**

A HR-TOF-AMS was deployed for online measurements of non-refractory (NR) PM$_{2.5}$ (with an inline PM$_{2.5}$ cyclone). A

detailed description of the instrument can be found elsewhere (Jayne et al., 2000; DeCarlo et al., 2006). The AMS recorded data with 1 min time resolution, of which 30 s was spient recording the ensemble mass spectrum (mass spectrum (MS) mode) and 30 s recording size-resolved mass spectra ("particle time-of-flight (ePToF) mode"). A Nafion dryer was used to dry the sampled air stream, which kept the relative humidity (RH) of air below 30 % within the AMS. Particles are continuously sampled (~0.8 L min$^{-1}$) through a 100 μm critical orifice and are focused by a recently developed PM$_{2.5}$ aerodynamic lens

(Williams et al., 2013). The particles impact on a heated tungsten surface (heated to 600°) at ~10$^{-7}$ Torr and the NR components are flash vaporized. The resulting gases are ionized by electron ionization (EI, ~70 eV) and the *m/z* values of the resulting fragments are determined by the TOF mass spectrometer. The AMS was calibrated for inlet flow, ionization efficiency (IE) at the beginning, middle and end of the campaign following standard protocols.

AMS data were analyzed in Igor Pro 6.36 using the SQUIRREL (version 1.57) and PIKA (1.16) analysis software (Donna

Sueper, ToF-AMS High Resolution Analysis Software). The collection efficiency (CE) was estimated using a composition-dependent collection efficiency (CDCE) algorithm (Middlebrook et al., 2012) implemented in SQUIRREL. A




CE=0.5 was assumed except in the case of strongly acidic aerosols, and high ammonium nitrate content where the approach by Middlebrook et al. (2012) was applied.

For ME-2 analysis, the input matrices consisted of the time series of fitted ions from high resolution mass spectral analysis, together with their corresponding uncertainties (Allan et al., 2003). According to the recommendations of Ulbrich et al. 2009, a minimum error value was added to the error matrix and ions were assessed and treated according to their signal-to-noise ratio (SNR) as follows: ions with an SNR less than 0.2 were excluded from ME-2 analysis, while those with an SNR between 0.2 and 2 were down-weighted by increasing their uncertainties by a factor of 2. Further, ions that were not independently fit but rather calculated from $CO_2^+$ were removed to avoid overweighting $CO_2^+$. Likewise, isotopes were not included in the matrices to avoid overweighting the parent ions. The source apportionment input matrices consisted of 257 ions between $m/z$ 12 and 120.

## 2.3 Source apportionment techniques

Source apportionment was performed on the organic AMS and EESI-TOF data using PMF as implemented by the multilinear engine (ME-2) (Paatero, 1997) and with model configuration and analysis executed via the SoFi (Source Finder, version 6.39) interface (Canonaco et al., 2013), coded in Igor Pro (WaveMetrics 6.37). PMF represents the input data matrix as a linear combination of characteristic factor profiles and their time-dependent contributions, which can be expressed in matrix notation as:

$$\mathbf{X} = \mathbf{G} \times \mathbf{F} + \mathbf{E} \tag{3}$$

The measured $\mathbf{X}$ is an $m \times n$ matrix, representing $m$ measurements of $n$ $m/z$. $\mathbf{G}$ and $\mathbf{F}$ are $m$ x $p$ and $p$ x $n$ matrices, respectively, where $p$ is the number of factors contained in a given model solution and is selected by the user.

Equation (3) is solved using a least squares algorithm that iteratively minimize the quantity $Q$ (Eq. 4), defined as the sum of the squared residuals weighted by their respective uncertainties, where the uncertainty may contain the measurements and model uncertainty:

$$Q = \sum_i \sum_j \left(\frac{e_{ij}}{\sigma_{ij}}\right)^2 \tag{4}$$

Here, $e_{ij}$ represents the residuals (elements of $\mathbf{E}$), with $i$ and $j$ denoting respectively the time and $m/z$ indices, and $\sigma_{ij}$ is the corresponding measurement uncertainty. Rotations are explored by using the $a$-value approach, here implemented by constraining one or more output factor profiles to resemble a selected source, improving source separation (Crippa et al., 2014; Canonaco et al., 2013). The $a$-value (ranging from 0 to 1) determines how much the constrained factor ($f_j$, solution) is allowed to vary from its anchor ($f_j$), as defined in Eq. (5).

$$f_{j,solution} = f_j \pm a \cdot f_j \tag{5}$$



Factors related to traffic and cooking were constrained for the AMS analysis, while a factor related to cigarette smoke was constrained for the EESI-TOF. Details are presented in Sect. 3.1 and 3.2.

## 2.4 Wind regression analysis

Wind regression analysis has been developed as a means of using meteorological and pollutant data to estimate the percent of

a given pollutant originating from a specific wind sector. This study utilizes the Sustained Wind Incidence Method (SWIM), a quantitative model that estimates the weighted pollutant concentrations and uncertainties from a given wind direction and speed (Henry et al., 2009; Olson et al., 2012). The expected concentration (E) of a pollutant for each wind direction/wind speed pair $(\theta, u)$ is calculated as a weighted average of the concentration data in a window around $(\theta, u)$ represented by smoothing parameters $\sigma$ and $h$ using a weighting function $K(\theta, u, \sigma, h) = K_1(\theta, \sigma) K_2(u, h)$, given by Eq (6):

$$E(C|\theta, u) = \frac{\sum_{i=1}^{N} K_1(\frac{\theta - W_i}{\sigma}) \cdot K_2(\frac{u - U_i}{h}) \cdot C_i}{\sum_{i=1}^{N} K_1(\frac{\theta - W_i}{\sigma}) \cdot K_2(\frac{u - U_i}{h})} \tag{6}$$

$$K_1(x) = \frac{1}{\sqrt{2\pi}} \cdot e^{-0.5 \cdot x^2}, \ -\infty < x < \infty \tag{7}$$

$$K_2(x) = 0.75 \cdot (1 - x^2), \ -1 < x < 1 \tag{8}$$

$$W_i = \frac{C_i U_i}{\max(C_i U_i)} \cdot \frac{\overline{(\sigma \theta_i)}}{\sigma \theta_i} \tag{9}$$

where $C_i$, $U_i$, and $W_i$ are the observed concentration of a particular pollutant, resultant wind speed and directional standard

deviation, respectively, $N$ is the total number of observations; $K_1$ (Eq (7)) and $K_2$ (Eq (8)) are smoothing Gaussian kernel and the Epanechnikov kernel, $\sigma$ and $h$ are smoothing parameters for wind direction and wind speed, respectively. The conditional probability of pollutant concentration (Eq (6)) is then weighted by the frequency of the wind using a joint probability of wind speed and wind direction, resulting in the following expression for the mean value of the pollutant concentration associated with winds from the sector defined by the intervals $U$ ($U = [u_1, u_2]$) and $\Theta$ ($\Theta = [\theta_1, \theta_2]$).

$$S(\Theta, U) = \int_{u_1}^{u_2} \int_{\theta_1}^{\theta_2} f(\theta, u) E(C|\theta, u) d\theta du \tag{10}$$

The joint probability of wind speed and wind direction ($f$) is calculated by using a kernel density, estimated as:

$$f(\theta, u) = \frac{1}{N\sigma h} \sum_{i=1}^{N} K_1\left(\frac{(\theta - W_i)}{\sigma}\right) K_2\left(\frac{(u - U_i)}{h}\right) \tag{11}$$

Calculations have been performed on Igor Pro with ZeFir package (Petit et al., 2017).

## 2.5 Identification of source-specific ions

To determine ions characteristic of individual factors (or groups of related factors), agglomerative hierarchical clustering was conducted on the EESI-TOF matrix of PMF profiles and standardizing data along the ions, clustering first along the columns



(producing row-clustered groups of factor), and then along the rows (producing the clustered ions to each group) in the matrix data. In hierarchical cluster analysis, a dendrogram, used to show relationships between members of a group, i.e., a family tree with the oldest common ancestor at the base and branches for various divisions of lineage was generated with the following steps (Matlab R2017b): (1) Calculate the distance by using Euclidean distance to find the similarity or dissimilarity between every ion and every pair of factors in our data set. (2) Link pairs of ions and factors that are in close proximity using the average linkage function. (3) Use the cluster function to prune branches off the bottom of hierarchical tree, and assign all the objects below each cut to a single cluster. Here, the clustergram function transforms the standardized values so that the mean is 0.

## 3 Results and Discussion

### 3.1 AMS source apportionment

The AMS PMF analysis yielded seven OA factors: hydrocarbon-like OA ($HOA_{AMS}$), cooking-related OA ($COA_{AMS}$), biomass burning OA ($BBOA_{AMS}$), two oxygenated OA factors ($OOA1_{AMS}$ and $OOA2_{AMS}$), nitrogen-containing OA ($NOA_{AMS}$), and a factor due to an isolated local event ($EVENT_{AMS}$). Factor mass spectra are shown in Fig. 1, while Fig. S1 shows factor time series, together with selected external tracers, and diurnal cycles which may be less convinced due to the short period of the measurement. Salient characteristics of these factors are discussed below; $HOA_{AMS}$, $COA_{AMS}$, $BBOA_{AMS}$, $OOA1_{AMS}$, and $OOA2_{AMS}$ are similar to factors frequently observed in other studies (Crippa et al., 2013a; Zhang et al., 2011; Young et al., 2016).

$HOA_{AMS}$ was constrained using a factor mass spectrum from Paris (Crippa et al., 2013b) and an *a*-value of 0.1, yielding a factor with a low O:C ratio (0.04) and high H:C ratio (1.8), consistent with a dominant contribution from aliphatic hydrocarbons. Strong signals from $C_xH_y^+$ ions are evident, especially $C_3H_5^+$, $C_3H_7^+$, $C_4H_7^+$, and $C_4H_9^+$ ions. Consistent with previous studies, the HOA mass spectrum is similar to vehicle emission studies (Zhang et al., 2005; Sun et al., 2012; Young et al., 2016).

The $COA_{AMS}$ mass spectrum is similar to primary cooking emissions (Crippa et al., 2013b) and exhibits a unique diurnal pattern peaking during lunch and dinner time. The $COA_{AMS}$ spectrum is characterized by a high ratio of $C_4H_7^+$:$C_4H_9^+$ ratio and a high fraction of $C_3H_3O^+$ and $C_4H_7^+$, consistent with $COA_{AMS}$ factors previously identified at urban locations (Crippa et al., 2013a; Ge et al., 2012; Mohr et al., 2012).

$BBOA_{AMS}$ has been identified as a significant source of aerosol in previous wintertime source apportionment studies in Switzerland and Central Europe (Lanz et al., 2008; Daellenbach et al., 2017). Similar to previous studies, $BBOA_{AMS}$ shows a high fraction $C_2H_4O_2^+$ at *m/z* 60 and $C_3H_5O_2^+$ at *m/z* 73 and explains most of the variation of these ions (77 %, 65 %, respectively). A strong diurnal trend is evident, with concentrations peaking overnight and decreasing during the day.

$OOA1_{AMS}$ and $OOA2_{AMS}$ mass spectra are characterized by dominant peaks at *m/z* 28 ($CO^+$), 44 ($CO_2^+$), similar to $OOA_{AMS}$ factors observed at other sites (Sun et al., 2011; Ng et al., 2010). The main difference between the $OOA1_{AMS}$ and $OOA2_{AMS}$ mass spectra is the relative contribution of $C_2H_3O^+$ compared to $CO_2^+$, with $C_2H_3O^+$ enhanced in $OOA1_{AMS}$. Also enhanced in





OOA1 are ions at *m/z* 39 ($C_3H_3^+$), 41 ($C_3H_5^+$), 55 ($C_4H_7^+$). OOA2 $_{AMS}$ shows a strong correlation with nitrate (R=0.82), which likely indicates a more volatile component. Further insight into the OOA$_{AMS}$ factors is obtained through the EESI analysis (Sect. 3.2).

NOA$_{AMS}$ exhibits a significantly higher N:C ratio (0.04) than the other factors and explains most of the organic nitrogen signal.

This factor includes a strong signal from $C_5H_{10}N^+$ signal (*m/z* 84), which is consistent with N-methyl-pyrrolidine which has previously been identified in AMS spectra as a tracer for cigarette smoke (Struckmeier et al., 2016). This ion is also observed in the EI mass spectra of nicotine (NIST, https://webbook.nist.gov/cgi/cbook.cgi?ID=C54115&Mask=200#Mass-Spec). However, other spectral features (e.g. the high $CO_2^+$ signal) are not typical of primary cigarette smoke and suggest a contribution from secondary formation processes. This interpretation is consistent with correlations of NOA$_{AMS}$ with

EESI-TOF factors, suggesting NOA$_{AMS}$ to be a mixed factor, as discussed in Sect. 3.2 and 3.4.

EVENT$_{AMS}$ factor is a special case in our study as the mass spectrum is dominated by *m/z* 15 ($CH_3^+$), 27 ($C_2H_3^+$), 31 ($CH_3O^+$), and 43 ($C_2H_3O^+$). The time series only contributes during two nights (28 January and 29 January) from 00:00 am to 07:00 am with the concentrations peaking at 3.8 µg m$^{-3}$ but being below 0.2 µg m$^{-3}$ for the rest of the study. No associations with any markers are evident.

**3.2 EESI-TOF source apportionment**

An 11-factor solution was selected as the best representation of the EESI-TOF data, with 5 factors attributed mostly to POA and 6 to SOA. The POA factors include cooking-related OA (COA$_{EESI}$), two less aged biomass burning factors (LABB1$_{EESI}$ and LABB2$_{EESI}$) which are mostly dominated by primary organic aerosol compounds, cigarette smoke-influenced OA (CS-OA$_{EESI}$), and a factor related to an isolated special event (EVENT$_{EESI}$). The SOA factors consist of 3 more aged biomass

burning factors dominated by secondary organic aerosol compounds and distinguished by mean daily temperature (MABB_LOW$_{EESI}$, MABB_HIGH$_{EESI}$, and MABB_TRANS$_{EESI}$, corresponding to low temperature, high temperature, and transition periods, respectively), two additional SOA factors lacking a clear attribution to biomass burning (SOA1$_{EESI}$ and SOA2$_{EESI}$), and nitrogen-containing SOA (NSOA$_{EESI}$). This solution was obtained by constraining the CS-OA$_{EESI}$ factor with an *a*-value of 0.1, and all other factors unconstrained. This constraining approach and the solution selection criteria are

discussed in Sect. 3.2.1, while the POA and SOA factors are discussed in Sect. 3.2.2 and 3.2.3, respectively. A detailed investigation of the factor mass spectra is presented in Sect. 3.4.

**3.2.1 Selection of PMF solution**

In selecting the PMF solution that best represents the EESI-TOF dataset, we considered both mathematical diagnostics (e.g. $Q/Q_{exp}$ and residuals) as a function of the number of factors, as well as the interpretability of the retrieved factors.

Interpretability was judged according to the following criteria:

   i. Correlation of the time series and diurnal patterns between the AMS and EESI-factors.





   ii. Comparison of factor profiles with mass spectra retrieved from less and more aged biomass burning exhaust from simulation chamber experiments at PSI (Bertrand et al., in prep.).

   iii. Similarities to EESI-TOF factor mass spectra retrieved from summer measurements at the same site in Zurich (Stefenelli et al., in prep.)

   iv. Identification of key ions in the factor profiles, including ions contributing a major fraction of the total factor signal, ions apportioned predominantly to a certain factor or related to a set of factors, and ions established in the literature as known tracers for specific sources/processes.

   v. Interpretation of the temporal behavior in terms of meteorological data, including temperature, solar radiation, and wind speed/direction.

For the EESI-TOF source apportionment, we considered unconstrained solutions from 7 to 20 factors (see Fig. S2a). Of these solutions, a 10-factor solution was found to best explain the data at a preliminary stage. This was preferred to lower-order solutions because all factors were interpretable according to the above criteria. Solutions with more factors lead to additional factors related to more aged biomass burning without obvious additional information. In addition, the investigation of $Q/Q_{exp}$ as a function of the number of factors (Fig. S2b) did not show any significant change with the increase of the $a$-value from 7 factors. Fig. S3 and Fig. S4 show the mass spectra and time series of the 8- to 11-factor solutions.

Nonetheless, the unconstrained 10-factor solution revealed evidence of factor mixing, as the cooking-related ($COA_{EESI}$) factor mass spectrum had a strong contribution from $m/z$ 163 ($C_{10}H_{15}N_2$, nicotine), which should rather be associated with cigarette smoke (Fig. S5). This suggests that at least one more factor remained to be resolved. For solutions with fifteen to twenty factors, a factor was retrieved with an MS dominated by nicotine and to which > 90% of nicotine was apportioned. We therefore constructed a profile (average from 15-20 factors) for this nicotine-containing factor (apportioned to cigarette smoke, i.e. $CS\text{-}OA_{EESI}$). This profile was then constrained in an 11-factor solution (based on the selection of a 10-factor unconstrained solution, as discussed above) using an $a$-value approach (from 0 to 1 with steps of 0.1, 0.1 was chosen finally). The main criterion of the constraint was the fraction of nicotine apportioned to the constrained factor. Also in our case, the $R^2$ (Pearson) for the correlations between the time series of the solutions was constructed with the final 11-factor solution. Based on these considerations, we concluded that the source apportionment solution with eleven factors was the optimal solution.

### 3.2.2 EESI-TOF Factors: Primary Organic Aerosols (POA)

Figure 2a shows the time series of the five EESI-TOF factors attributed to primary organic aerosol: $COA_{EESI}$, $LABB1_{EESI}$, $LABB2_{EESI}$, $CS\text{-}OA_{EESI}$, and $EVENT_{EESI}$. Also shown are relevant ancillary measurements, including AMS PMF factors and meteorological parameters. Figure 2b shows the corresponding factor mass spectra, colored by the number of nitrogen atoms. A discussion of each factor follows. Figure 3a shows the diurnal patterns of the $LABB_{EESI}$ factors, as well as $COA_{EESI}$ and $COA_{AMS}$.

***Less aged biomass burning ($LABB1_{EESI}$ and $LABB2_{EESI}$)***



The LABB factors are both enhanced at night, consistent with domestic heating activities. Considering the full campaign time series (Fig. 2a), this repeating pattern, opposed to solar radiation, is evident for $LABB1_{EESI}$, while the time series of $LABB2_{EESI}$ is driven by intense events (~6.5 times higher than $LABB1_{EESI}$) during two nights: from 18:00 on 27 January to 08:00 on 28 January, and from 18:00 on 28 January to 08:00 on 29 January. As shown in Fig. 2b, both factor profiles are dominated by

$C_6H_{10}O_5$ and $C_8H_{12}O_6$. $C_6H_{10}O_5$ is attributed primarily to levoglucosan, which is a well-established tracer for biomass burning. $C_8H_{12}O_6$ may represent hydroperoxides from the oxidation (reaction with OH radical during daytime) of phenolic compounds (Yee et al., 2013; Nakao et al., 2011). The mass spectrum features of both factors are very similar to less aged biomass burning emissions measured directly from a domestic biomass combustion appliance in the PSI smog chamber (Bertrand et al., in prep.). Figures 4a and 4b show Van Krevelen plots (i.e., atomic ratios H:C as a function of O:C) for $LABB1_{EESI}$ and $LABB2_{EESI}$,

respectively, with points colored by the number of atoms and sized by the fraction of each ion apportioned to the respective factor. Both $LABB1_{EESI}$ and $LABB2_{EESI}$ are dominated by ions with low H:C (1.04) and low O:C (0.35, excluding the sugars $C_6H_{10}O_5$ and $C_8H_{12}O_6$, which exhibit high variability, Table S1), suggesting a strong contribution from primary or slightly aged aromatics. Fig. S6 compares the $BBOA_{AMS}$ factor (Fig. 2a) with $LABB1_{EESI}$, $LABB2_{EESI}$, and the sum of $LABB1_{EESI}$ + $LABB2_{EESI}$, with $R^2$ 0.35, 0.63, and 0.68, respectively. It's worth noting that something "different" happens during the first

part of the study (25 January to 27 January) which we will discuss later. The correlation of $BBOA_{AMS}$ with either $LABB2_{EESI}$ or $LABB1_{EESI}+LABB2_{EESI}$ is rather high at night ($R^2$=0.35 to 0.68), while the $LABB_{EESI}$ factors are consistently lower than $BBOA_{AMS}$ during the day. We assign the high correlation of $LABB2_{EESI}$ with $BBOA_{AMS}$ to the high abundance of levoglucosan which drives the variation in $f60$ in the AMS. Some specific features of $BBOA_{AMS}$ do not appear in any LABB factor because less aged and more aged biomass burning OA are not unambiguously separated in the AMS.

### Cooking-related OA (COA_{EESI})

The $COA_{EESI}$ and $COA_{AMS}$ factor time series are strongly correlated (R=0.88), as shown in Fig. 2a. The diurnal variation of the $COA_{EESI}$ is also similar to $COA_{AMS}$, with strong peaks at lunch and dinner time (Fig. 3a). In addition to this diurnal pattern, both $COA_{EESI}$ and $COA_{AMS}$ are significantly elevated during two periods: from18:00 on 27 January to 01:00 on 28 January

(Friday night), and from 18:00 on 28 January to 01:00 on 29 January (Saturday night). These periods occur on the same evening as the unknown special event giving rise to the $EVENT_{AMS}$ and $EVENT_{EESI}$ factors, but are slightly offset in time, with the COA factors peaking approximately four hours earlier. The distinct contribution from the $COA_{EESI}$ factor is due in part to the location of several restaurants within a 100-m radius, including one adjacent to the site.

As shown in Fig. 2b, the $COA_{EESI}$ mass spectrum is unique in having most of the mass at ions with higher $m/z$. Several of the

dominant ions can be attributed to fatty acids and alcohols, which are associated with cooking emissions and oils. For example, $C_{13}H_{22}O_4$ (dibutyl itaconate), $C_{16}H_{30}O_3$ (2-oxo-tetredecanoic acid), and $C_{18}H_{34}O_3$ (ricinoleic acid), are prominent, and contributed 0.89 %, 1.7 %, and 2.0 %, respectively, of the total mass spectrum. Figure 3b shows a Van Krevelen plot of the $COA_{EESI}$ factor mass spectrum, with points sized by the fraction of each ion apportioned to $COA_{EESI}$ and colored by the number of carbon atoms. The dominant contribution of ions with higher carbon number (C13-C25) and high H:C ratio (greater than 1.5)





but low O:C ratio (below 0.2) indicates that these ions are more consistent with fatty acids or alcohols rather than aromatic-derived ions.

### Special event (Event_EESI)

The time series of EVENT$_{EESI}$ is highly correlated with EVENT$_{AMS}$ (R=0.99, Fig. 2a). Both factors are near-zero except for two intense events beginning at approximately midnight and lasting till the early morning on 28 and 29 January, supporting the hypothesis of a unique event as opposed to variation in BBOA. The nature of this event is not known, and no human activities in the immediate vicinity of the sampling inlet were evident by inspection of the on-site camera. The EESI-TOF factor mass spectrum is dominated by an ion at $m/z$ 174.08, tentatively assigned to $C_8H_{11}N_2O$. However, the EESI-TOF does not provide structural information and to our knowledge no compound with this formula has been reported as a major constituent of an atmospheric emission source, preventing its use as a diagnostic tracer. Other significant ions are $C_8H_{12}O_4$ and $C_8H_{18}O_5$. The $C_8H_{12}O_4$ ion likely represents 1,2-cyclohexane dicarboxylic acid diisononyl ester, a plasticizer for the manufacture of food packaging, belonging to the group of aliphatic esters from a chemical point of view. This indicates that the source may be from food plastic burning in a nearby restaurant.

### Cigarette smoke-influenced OA (CS-OA_EESI)

Cigarette smoke-influenced OA (CS-OA$_{EESI}$) is a constrained factor, based on a reference profile retrieved from higher-order PMF solutions as described in Sect. 3.2.1. The mass spectrum of CS-OA$_{EESI}$ is dominated by the $C_{10}H_{14}N_2H^+$ ion (Fig. 2b). This ion is the only ion (out of 892 ions) that does not appear as an adduct with $Na^+$. Instead, the observed molecular formula corresponds to that of nicotine with an extra hydrogen. As a reduced nitrogen compound, nicotine likely forms a stable ion by abstracting a hydrogen from water, leading to the observed cation. However, the time series and the mass flux of this ion should be interpreted with caution, because it is formed by a different ionization pathway than the majority of the spectrum, its relative sensitivity may be significantly different from that of the other ions. Additionally, we have not characterized such non-Na-adducts in terms of ion suppression or matrix effects and cannot rule out a nonlinear response to mass. However, the comparison of the CS-OA$_{EESI}$ factor with AMS PMF results and individual ions discussed below suggest that such nonlinear effects are not significant.

Oxidized organic nitrogen species such as $C_xH_yN_1O_z$ (34.9 %) and $C_xH_yN_2O_z$ (6.8 %) are also significant in the CS-OA$_{EESI}$ factor, as shown in Fig. 2b and Fig. 5a. CS-OA$_{EESI}$ is only slightly oxygenated (O:C=0.31) and has an H:C ratio of approximately 1.51 (Table S1). The CS-OA$_{EESI}$ time series exhibits two large evening peaks (27 Jan and 28 Jan). These peaks are likely associated with cigarette smoking outside the nearby restaurants. A high correlation is observed between the time series of CS-OA$_{EESI}$ and the AMS $C_5H_{10}N^+$ ion (R=0.91, Fig. 5b), which has been proposed as a tracer for nicotine (Struckmeier et al., 2016).

### 3.2.3 EESI-TOF Factors: Secondary Organic Aerosols

Here we discuss the EESI-TOF SOA factors in three groups: (1) more aged wood-burning related (MABB_LOW$_{EESI}$, MABB_TRANS$_{EESI}$, and MABB_HIGH$_{EESI}$); (2) non-source-specific SOA (SOA1$_{EESI}$ and SOA2$_{EESI}$); and (3) high nitrogen



content (NSOA$_{EESI}$). Factor mass spectra for these factors are shown in Fig. 6a, with the spectra colored by the number of N atoms and normalized such that the sum of the peaks in each spectrum is 1. Figure 6b shows a stacked time series of all 6 EESI-TOF SOA factors, such that the sum of the stacked plot represents the total EESI-TOF mass flux attributed to SOA. For comparison, the time series of the estimated AMS SOA is shown, calculated as OOA1$_{AMS}$ + OOA2$_{AMS}$. NSOA$_{AMS}$ is excluded

from this calculation due to the contribution from primary cigarette smoke discussed above. The total EESI-TOF SOA and AMS SOA estimates are in general well-correlated (R=0.90), even though the EESI-TOF mass flux is proportionally lower during the first few days of the study.

*More aged biomass burning-related factors (MABB_LOW$_{EESI}$, MABB_TRANS$_{EESI}$, and MABB_HIGH$_{EESI}$)*

Three more aged biomass burning (MABB) factors are identified in this study: MABB_LOW$_{EESI}$, MABB_TRANS$_{EESI}$, and

MABB_HIGH$_{EESI}$. Each MABB$_{EESI}$ factor is enhanced relative to the others during a different part of the campaign, which correspond to both changes in the daily temperature cycle and wind direction. As shown in Fig. 6b, the coldest part of the study occurs from 25 to 27 January (mean -5.4°, min -6.4°, max -2.2°), period 1. During this period, MABB_LOW$_{EESI}$ contributes 84 % of the total MABB (MABB_LOW$_{EESI}$/(MABB_LOW$_{EESI}$ + MABB_TRANS$_{EESI}$ + MABB_HIGH$_{EESI}$)). From 27 to 29 January, Period 2, temperature increases (mean 1.4°, min -2.2°, max 7.4°), and the MABB_TRANS$_{EESI}$ factor constitutes the

dominant MABB$_{EESI}$ fraction (65 %). Period 3, from 29 January to the campaign end on 4 February, corresponds to higher temperatures (mean 5.7°, min 0.8°, max 8.7°), and the MABB$_{EESI}$ fraction is dominated by MABB_HIGH$_{EESI}$ (90 %) until a substantial precipitation event beginning on 31 January, after which relatively clean air is observed for the remainder of the campaign. Figure 7 shows the source-specific wind sectors determined by SWIM (see Sect. 2.4) for the three MABB factors. This analysis assigns the three factors to distinct wind vectors: NNE for MABB_LOW$_{EESI}$, NNW for MABB_TRANS$_{EESI}$, and

SE for MABB_HIGH$_{EESI}$. Because each factor is predominantly observed during a single time period, it is difficult to assess the relative importance of temperature vs. source region for these three factors.

As shown in Fig. 6a, all three MABB$_{EESI}$ factor mass spectra are qualitatively similar, with many of the same ions enhanced. These spectra are also similar to the mass spectrum of aged biomass burning emissions retrieved from smog chamber experiment (Bertrand et al., in prep.). For both the MABB$_{EESI}$ and chamber spectra, the major ions, C$_7$H$_{10}$O$_5$, C$_9$H$_{14}$O$_4$,

C$_8$H$_{12}$O$_6$, are in common. The main difference between the EESI-TOF factors and the chamber mass spectrum is that the chamber data show a higher fraction of signal at lower *m/z*. This is likely due to the higher concentrations used during the chamber experiments, causing increased partitioning of semi-volatile compounds to the particle phase. MABB_LOW$_{EESI}$ also exhibits somehow enhanced intensities at lower *m/z* compared to the other MABB$_{EESI}$ factors. As MABB_LOW$_{EESI}$ is dominant during the coldest period 1, the MABB_LOW$_{EESI}$ factor is possibly separated from the other MABB$_{EESI}$ factors due

to partitioning of semi-volatile material to the particle phase due to colder temperatures.

Further insight into the composition trends across the MABB$_{EESI}$ factors is obtained through Fig. 8 which represents the three MABB$_{EESI}$ mass spectra as the carbon oxidation state (OS$_c$) (Kroll et al., 2011) of each ion as a function of the carbon number ($n_c$). Data points are colored by the H:C ratio and sized by the fraction of each ion apportioned to the designated factor. The figure shows that MABB_LOW$_{EESI}$ is enhanced in ions with low $n_c$, consistent with condensation of semi-volatile OA (C$_5$H$_6$O$_4$,





$C_8H_6O_4$, $C_5H_8O_7$) at low temperature. Otherwise, all three $MABB_{EESI}$ factors are rather similar. Figure 8 also shows the $OS_c$ of non-$MABB_{EESI}$ (weighted average of $SOA1_{EESI}$ + $SOA2_{EESI}$) and $LABB_{EESI}$ (weighted average of $LABB1_{EESI}$ and $LABB2_{EESI}$) factors. Obviously, the non-$MABB_{EESI}$ and $LABB_{EESI}$ factors are less oxidized than the MABB factors, with lower $OS_c$.

### *Other SOA factors ($SOA1_{EESI}$ and $SOA2_{EESI}$)*

The mass spectra of $SOA1_{EESI}$ and $SOA2_{EESI}$ are qualitatively similar to factors retrieved from PMF analysis of EESI-TOF data from Zurich during summer, when monoterpenes are the dominant SOA precursors (Stefenelli et al, in prep.). Major ions include $C_8H_{12}O_4$, $C_9H_{14}O_4$, $C_{10}H_{16}O_4$, $C_{10}H_{18}O_4$, $C_{10}H_{16}O_5$ and $C_{10}H_{16}O_2$, $C_{10}H_{16}O_3$, $C_{10}H_{18}O_4$, separately. In contrast to the $MABB_{EESI}$ factors, the $SOA_{EESI}$ factors have a negligible contribution from levoglucosan ($C_6H_{10}O_5$). Approximately 57 % of the total C10 ion signal is apportioned to the $SOA_{EESI}$ factors. Figures 9a and 9b show the atomic ratio of H:C as a function of

O:C for the two $SOA_{EESI}$ factors. These H:C ratios are higher than typically observed from the oxidation of aromatic emissions and are instead consistent with monoterpene oxidation. The Van Krevelen plots show clear differences between these two factors, $SOA2_{EESI}$ is less oxygenated than $SOA1_{EESI}$ with lower O:C ratio and lower H:C ratio. The time series of $SOA1_{EESI}$ shows a higher contribution during the period 1, 2, while $SOA2_{EESI}$ has a more regular cycle contribution during daytime (Fig. 6b). Since we have clear evidence that these EESI-retrieved factors are related to secondary organic aerosol we call them

$SOA_{EESI}$, in contrast to the $OOA_{AMS}$ factors, where this evidence is less clear.

### *Nitrogen-containing SOA factor ($NSOA_{EESI}$)*

As mentioned in Sect. 3.1, the EESI-TOF source apportionment also resolves a nitrogen-containing SOA factor ($NSOA_{EESI}$). $NSOA_{EESI}$ is dominated by highly oxygenated organonitrate molecules, including $C_8H_{13}NO_5$, $C_{10}H_{15}NO_6$, $C_{10}H_{19}NO_8$. Ions like $C_6H_{10}O_5$, $C_{10}H_{16}O_2$ and $C_8H_{12}O_6$ are comprising another fraction of the $NSOA_{EESI}$ signal, but are not unique to the

$NSOA_{EESI}$ factor and rather spread over many other factors. The significant contribution of organonitrates results in an N:C ratio (0.05) and suggests a secondary origin for this factor. Therefore, we call it $NSOA_{EESI}$, in contrast to $NOA_{AMS}$ for which the primary/secondary origin is less certain. The time series of the factor is quite unique, shows maximum mass flux at the end of this campaign with the highest peak at night (3-4 February), and a smaller peak during the night of 28 to 29 February. Figure S7 shows a comparison of the $NSOA_{EESI}$ time series and $CS-OA_{EESI}$ time series with the CHON fragments from the

EESI and CHN ions from the AMS, respectively. The group of AMS_CHON fragments shows the same temporal variation as the $NSOA_{EESI}$ factor (Fig. S7) while the AMS_CHN group is more correlated to the primary organic group.

### 3.3 Analysis of marker ions

Laboratory, as well as offline and semi-continuous field studies have identified a number of tracer molecules that are useful for the investigation of primary and secondary OA from various sources, including biomass burning. The real-time and in situ

measurement of these compounds is a novel feature of the EESI-TOF, and their apportionment gives further insight into the nature of the factors described above. Here we investigate the apportionment of eight compounds: levoglucosan (1,6-anhydro-β-glucopyranose, $C_6H_{10}O_5$), methyl-nitrocatechol ($C_7H_7NO_4$), syringic acid ($C_9H_{10}O_5$), vanillic acid ($C_8H_8O_4$), phthalic acid ($C_8H_6O_4$), glutaconic acid ($C_5H_6O_4$), tetrahydroxy toluene ($C_7H_8O_4$) and pentahydroxy toluene ($C_7H_{10}O_5$). Figure



10a shows a stacked time series of the mass flux of these compounds representing the contribution of each EESI-TOF PMF factor to the total mass flux (assuming no significant conformational isomers). Levoglucosan, which is derived from the pyrolysis of cellulose and hemicellulose, is commonly used as an indicator for the presence of primary aerosols originating from biomass combustion (Fine et al., 2001). Figure 10b shows that levoglucosan appears in both POA (total contribution,

62 %, mostly from LABB1$_{EESI}$ (22 %) and LABB2$_{EESI}$ (37 %), and minor contributions by COA$_{EESI}$, CS-OA$_{EESI}$, and EVENT$_{EESI}$) and SOA (total contribution, 38 %, of which 36 % related to the sum of MABB_HIGH$_{EESI}$, MABB_TRANS$_{EESI}$, MABB_LOW$_{EESI}$, plus minor contributions from NSOA$_{EESI}$). Due to the high biomass burning emission background and the lifetime of levoglucosan, it is inevitable to find a contribution of levoglucosan in the MABB factor, which is consistent with our aged biomass burning discussion above. In contrast, nitrocatechol ($C_7H_7NO_4$) has been established as a secondary species

originating from the oxidation of biomass burning (Iinuma et al., 2010; Finewax et al., 2018). Here 86 % of nitrocatechol is apportioned to the less aged (49 %) and more aged (37 %) biomass burning factors. Syringic acid and vanillic acid are phenolic acids derived from the oxidation of lignin decomposition products (He et al., 2018), which in turn are a major component of biomass combustion emissions, and are apportioned primarily to the MABB$_{EESI}$ factors (68 % for syringic acid and 78 % for vanillic acid).

Phthalic acid ($C_8H_6O_4$) and glutaconic acid ($C_5H_6O_4$) are apportioned to the SOA factors (91 % and 94 % in total, respectively), with main contributions from the MABB$_{EESI}$ factors and in particular the MABB_LOW$_{EESI}$ factor (53 % and 59 %, respectively). These dicarboxylic acids are ubiquitous water-soluble organic compounds which have been detected in a variety of aerosol samples, and originate from the combustion of biomass burning and fossil fuels, as well as from biogenic emission and photo-oxidation of organic gases. For example, phthalic acid has been identified based on field measurements, as a tracer

of naphthalene oxidation (Kleindienst et al., 2012) or oxidation products from PAHs (Chan et al., 2009), and is also consistently found in combustion products of lignin, which is likely to explain the contribution in the MABB factors (Fu et al., 2010; Wang et al., 2007).

Tetrahydroxy toluene ($C_7H_8O_4$) and pentahydroxy toluene ($C_7H_8O_5$) are apportioned mainly to secondary factors (85.2 % and 78.6 %, respectively). Tetrahydroxy toluene and pentahydroxy toluene have been detected as dominant products both in the

particle phase and gas phase under low-NO oxidation of toluene (Nakao et al., 2012; Schwantes et al., 2017). The *o*-cresol oxidation mechanism for tetrahydroxy toluene and pentahydroxy toluene is found in MCM v3.3.1, based on Olariu et al. (2002). This formation indicates that these two low-volatility ions are indeed secondary organic compounds, consistent with our results shown in Fig. 10. In addition, the temporal variation of the pentahydroxy toluene contribution is consistent with the one of tetrahydroxy toluene except for the EVENT$_{EESI}$ factor, which may indicate that during this night event an isomer of

pentahydroxy toluene was present.

### 3.4 EESI-TOF cluster analysis

As evidenced from the previous section and Figs. 2 and 6, many of the dominant ions in the EESI-TOF PMF analysis are shared by multiple factors. Here, we utilize a cluster analysis to identify ions unique or nearly unique to a single factor or group



of factors. As discussed in Sect. 2.5, hierarchical agglomerative clustering is performed separately on the set of all EESI-TOF ions and all EESI-TOF factors. Figure 11 shows the resulting dendrogram of the ions and factors along the vertical and horizontal axes, respectively; the ion dendrogram is colored subjectively to guide the eye. Comparison of the ions to the factors yields a matrix, also shown in Fig. 11, which is colored by the z-score, with brown colors denoting high correlation. In this

representation, an ion unique to a given factor is brown for one and only one rectangle in the horizontal dimension.

The factor dendrogram resolves three main groups, (1) more aged biomass burning factors (MABB_LOW$_{EESI}$, MABB_TRANS$_{EESI}$ and MABB_HIGH$_{EESI}$), (2) less aged biomass burning factors (LABB1$_{EESI}$ and LABB2$_{EESI}$), and (3) the cooking-related OA and cigarette smoking OA factors. This grouping is consistent with our interpretation of these factors, as discussed in the previous section. Ions are clustered to different groups using the standardized values. In each factor, there are

distinguished molecules (lists of the specific ions (standardized value above 1.5) for each factor is shown in Table S2).

For several of the factors, the uniquely assigned ions exhibit systematic patterns contributing to the identification or deconvolution of the factors. Figure 12a shows the mass defect, defined as the exact $m/z$ minus the nearest integer $m/z$, as a function of $m/z$ for the uniquely assigned ions for the five POA$_{EESI}$ factors. Figure 12b shows the equivalent plot for the three MABB$_{EESI}$ factors and SOA1$_{EESI}$ (SOA2$_{EESI}$ and NSOA$_{EESI}$ have a high degree of scatter and are omitted to avoid masking

trends in the other secondary factors). The displayed factors exhibit linear correlations or tight clusters of points; all factors are shown independently in Fig. S8). The main feature in Fig. 12a is that LABB1$_{EESI}$ and LABB2$_{EESI}$ have a lower mass defect and shallower slope than COA$_{EESI}$ and CS-OA$_{EESI}$, consistent with increased aromaticity. Indeed, the slopes of the two LABB factors are nearly identical. The slopes are $(4.9\pm0.4)\cdot10^{-4}$, $(5.9\pm0.6)\cdot10^{-4}$, $(8\pm0.5)\cdot10^{-4}$ and $(8\pm0.3)\cdot10^{-4}$ for LABB1$_{EESI}$, LABB2$_{EESI}$, COA$_{EESI}$ and CS-OA$_{EESI}$, respectively. The slopes of the two LABB factors as well as those of COA$_{EESI}$ and CS-OA$_{EESI}$

are very similar to each other and are consistent with CH addition for the former (i.e. $C_{10+x}H_{14+x}O_{4-5}$, theoretical slope $6\cdot10^{-4}$), and CH$_2$ addition for the latter (i.e. $C_{10+x}H_{20+2x}O_{3-5}$ for COA$_{EESI}$ and $C_{10+y}H_{15+2y}NO_{3-5}$ for CS-OA$_{EESI}$ as nearly every CS-OA-specific ion contains a single N atom, theoretical slope $1.1\cdot10^{-3}$).

As shown in Fig.12b, the mass defect of SOA1$_{EESI}$ is much higher than those of three MABB factors and its slope ($0.9\cdot10^{-3}$) is consistent with the addition of CHO functionality (theoretical slope $= 0.1\cdot10^{-2}$). Due to the high variability of the slopes of the

MABB$_{EESI}$ factors, it is not possible to identify the added functionalities. Both mass defect and slope are higher for MABB_LOW$_{EESI}$ than for MABB_HIGH$_{EESI}$, which is consistent with our discussion in Sect. 3.2.3, assuming that the organics of the MABB_LOW$_{EESI}$ factor are more oxidized than those of the MABB_HIGH$_{EESI}$ factors. The LABB and MABB factors have similar slopes, despite different ion lists. In addition, the MABB intercepts are more positive than those of LABB, consistent with the higher oxidation state shown above.

**3.5 Comparison of AMS and EESI-TOF**

Fig. 13a shows the sum of the mass flux of the ions measured by the EESI-TOF as a function of the OA concentration measured by the AMS, with the points colored by date and time. We apply no ion-dependent sensitivity corrections for the



EESI-TOF, although ion-by-ion differences are known to exist (Lopez-Hilfiker et al., in prep.). Note that the AMS signal includes the minor OA source, HOA$_{AMS}$, which is mostly insoluble in the electrospray droplets and thus expected to be basically undetectable by the EESI-TOF. Nevertheless, the two instruments are well-correlated (R=0.94). The strong correlation in Fig. 13a suggests that the overall EESI-TOF sensitivity to OA does not vary significantly throughout the study,

and therefore it is unlikely that the major individual EESI-TOF PMF factors (which describe the compositional variability) have dramatically different response factors. We therefore interpret the EESI-TOF PMF results without correction of the data for factor-specific sensitivities. Several features are evident from dependence of the sensitivity on the mass flux of levoglucosan (Fig. 13b) which may explain the discrepancy in the first part of the campaign (period 1) vs. the rest of the campaign. An SOA-dominated period with low levoglucosan concentration (red line) including the early campaign data

exhibits a lower sensitivity than during a period with higher levoglucosan concentrations (black line) including the big mid-campaign events. Figures 13c and 13d respectively show the O:C and H:C atomic ratios for the EESI-TOF as a function of those for the AMS. Here again no ion-dependent sensitivity corrections are applied. The EESI-TOF and AMS O:C ratios are correlated (R=0.62), however, the O:C ratios estimated by the EESI-TOF are systematically higher than those measured by the AMS. For H:C ratios, we do not observe a correlation. The EESI-TOF values are scattered around approximately 1.56,

independent of the AMS H:C ratios which vary between 1.11 and 1.44. The cause for this discrepancy is not yet understood but may be related to differences in ion relative sensitivity (Bertrand et al., in prep.).

Fig. 14 shows the stacked time series of the EESI-TOF PMF factors (together with total AMS OA concentration) and of the AMS PMF factors. Also shown are pie charts denoting the mean OA PMF composition over the entire campaign from the EESI-TOF and AMS data. Despite uncertainties in the definition and resolution of primary vs. more aged biomass burning, the

AMS and EESI-TOF are in relatively good agreement with respect to the total POA and SOA fractions. The SOA factors comprise 58.8 % of the mass flux for the EESI-TOF and 69.4 % of the mass for the AMS. The agreement may in fact be better than these values indicate: as noted above the NOA$_{AMS}$ factor, comprising 17.9 % of the mass and fully associated to SOA in our solution, is likely composed of both POA (derived from cigarette smoke, as resolved in CS-OA$_{EESI}$) and SOA (from organonitrate-containing SOA, as resolved in NSOA$_{EESI}$), resulting in a low total POA fraction in the AMS solution. Since

both CS-OA$_{EESI}$ and NSOA$_{EESI}$ are enriched with the nitrogen-containing ions, we compare in Fig. 15, the O:C and N:C ratios for these two factors, where the size of the colored star and circle corresponds to the H:C ratio. A distinct separation between CS-OA$_{EESI}$ and NSOA$_{EESI}$ is evident due to a significantly higher O:C ratio for a given N:C ratio, i.e., higher degree of oxygenation for the NSOA$_{EESI}$ factor and a higher abundance of organic-nitrate molecules in the NSOA$_{EESI}$ factor. Moreover, this separation was not possible for AMS PMF.

Both AMS and EESI-TOF factors stacked time series (Fig. 14) show clearly that biomass burning is dominated by secondary fractions early in the campaign, mixed fractions in the middle of the campaign, and a primarily fractions late in the campaign. As discussed in Sect 3.2.2, BBOA$_{AMS}$ is a mixture of primary and secondary ions, and OOA$_{AMS}$ is a mixture of biomass burning fragments and background SOA fragments from photochemistry production ions. Although the fraction of OOA comprised more than 50 % percent of total OA (Fig. 14), it is hard to define how much of AMS OOA is WB-related as a





function of time. The EESI-TOF separates the biomass burning factors into LABB$_{EESI}$ and MABB$_{EESI}$ and splits the background SOA factors into separate factors, which provides evidence that biomass burning is the single most important contributor to the organic aerosol at the measurement site during winter.

**4 Conclusions**

Real-time, near-molecular level measurements of OA composition were performed during winter in Zurich using a novel extractive electrospray ionization time-of-flight mass spectrometer (EESI-TOF). The lack of thermal decomposition or ionization-induced fragmentation in the EESI-TOF provides an improved description of SOA in particular, facilitating SOA source identification by PMF. We retrieve eleven factors, of which 5 are dominated by POA and 6 by SOA. The POA factors

included cooking-influenced OA (COA$_{EESI}$, which strongly correlates with an equivalent AMS factor), cigarette smoke-influenced OA (CS-OA$_{EESI}$, characterized by a strong contribution from nicotine) and a special event also captured by the AMS. Two less aged biomass burning factors are also resolved. Of the six SOA factors, three are clearly related to biomass burning and are distinguished by temperature and possibly wind direction. We also observe two SOA factors with no clear biomass burning signatures, one of which closely resembles monoterpene oxidation. Finally, we observe a minor factor with a

high organonitrate fraction.

We performed cluster analysis of the EESI-TOF ions followed by correlation with the resolved factors, which identifies groups of ions characteristic of each factor. These characteristic ions represent potential tracers for future studies; they indicate strong aromatic influence in both primary and more aged biomass burning, and support the primary/secondary assignment of biomass burning-influenced factors.

The increased chemical specificity of the EESI-TOF allows for additional, meaningful factors to be resolved relative to the AMS. Comparisons of bulk measurements, as well as of individual factors or groups of factors between the EESI-TOF and AMS indicate good agreement. This suggests that, despite significant uncertainties in the relative response factors of individual ions measured by the EESI-TOF, responses at the level of the PMF factors are relatively similar, with the main differences resulting from the high sensitivity to levoglucosan in the EESI. As a result, the EESI-TOF represents a promising new

approach for source apportionment and atmospheric composition studies.

**Author Contribution.**

LQ was the main author. LQ, GS, VP, YT, and CH conducted the field campaign. MD, XG, JS, AP, and UB were the supervisors. All contributed to the corrections of the paper.





**Competing interests.**

The authors declare that they have no conflict of interest.

**Acknowledgements.**

This study was funded by the Swiss National Science Foundation (starting grant BSSGI0_155846), the National Natural
Science Foundation of China (grant No. 91543115, 21577065), and the International ST Cooperation Program of China
(2014DFA90780). We acknowledge the support by the Federal Office for the Environment. Mao Xiao is acknowledged for
useful discussions. The authors gratefully acknowledge technical and logistical support from R. Richter (PSI).

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



**Figures**



5   **Fig. 1. Factor profiles for the 7-factor AMS PMF solution. HOA$_{AMS}$ is constrained by *a*-value 0.1. The total signal of each factor is normalized to unity. (HOA$_{AMS}$: Hydrocarbon OA, COA$_{AMS}$: Cooking-related OA, BBOA$_{AMS}$: Biomass burning OA, OOA$_{AMS}$: Oxygenated OA, NOA$_{AMS}$: Nitrogen containing OA, EVENT$_{AMS}$, an isolated local event)**





**Fig. 2. Time series of the POA factors retrieved from EESI-TOF PMF analysis, along with ancillary data (a), and corresponding factor profiles (b). For all y-axes, EESI-TOF data are shown in mass flux (ag/s), AMS data are shown in μg m⁻³, and other units are given. Factor profiles are molecular weighted and are normalized such that the sum of each profile is 1.**





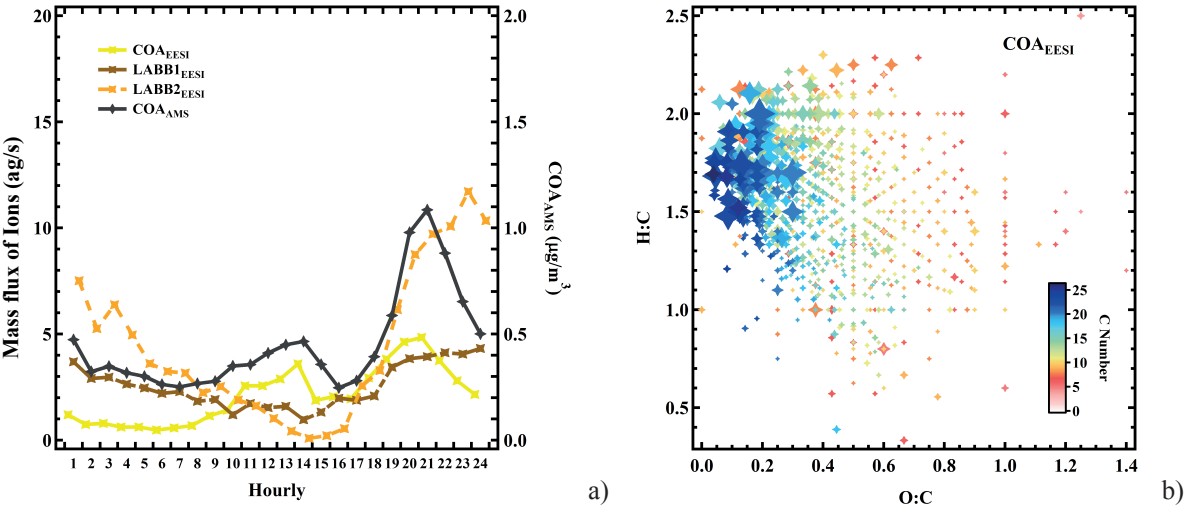

**Fig. 3. a) Diurnal cycles of the EESI-TOF less aged biomass burning and cooking factors, together with AMS cooking. b) Van**
**5 Krevelen plot (atomic H:C vs. O:C of each ion) for the COA$_{EESI}$ factor mass spectrum, with points sized by the fraction of each ion**
**apportioned to COA$_{EESI}$ and colored by number of carbon atoms.**

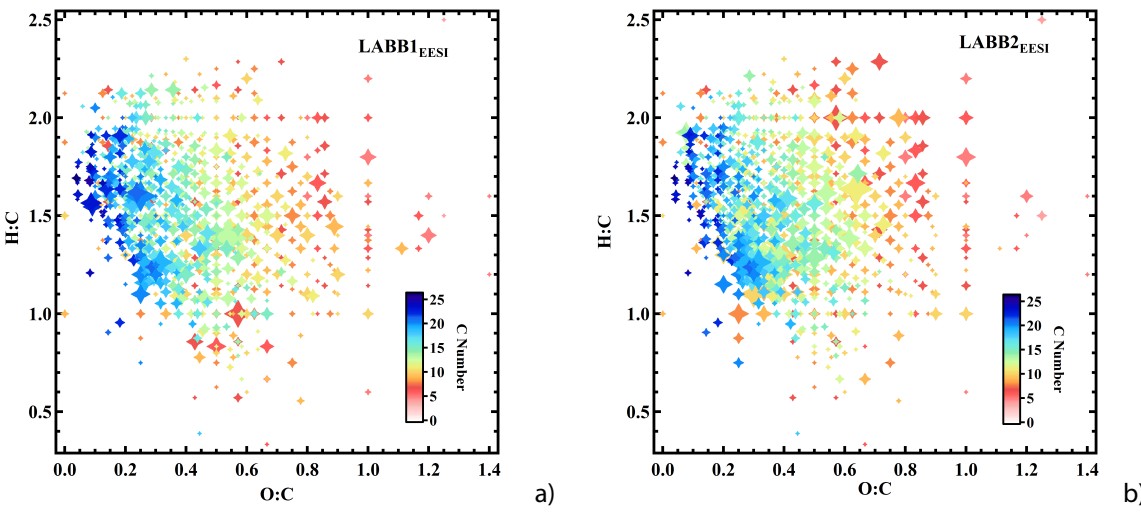

**Fig. 4. a), b) Van Krevelen plot (atomic H:C vs. O:C ratio) of the LABB1$_{EESI}$ and LABB2$_{EESI}$ factor mass spectra. Points are sized**
**10 by the fraction of each ion apportioned to LABB1$_{EESI}$ and LABB2$_{EESI}$ and colored by number of carbon atoms.**





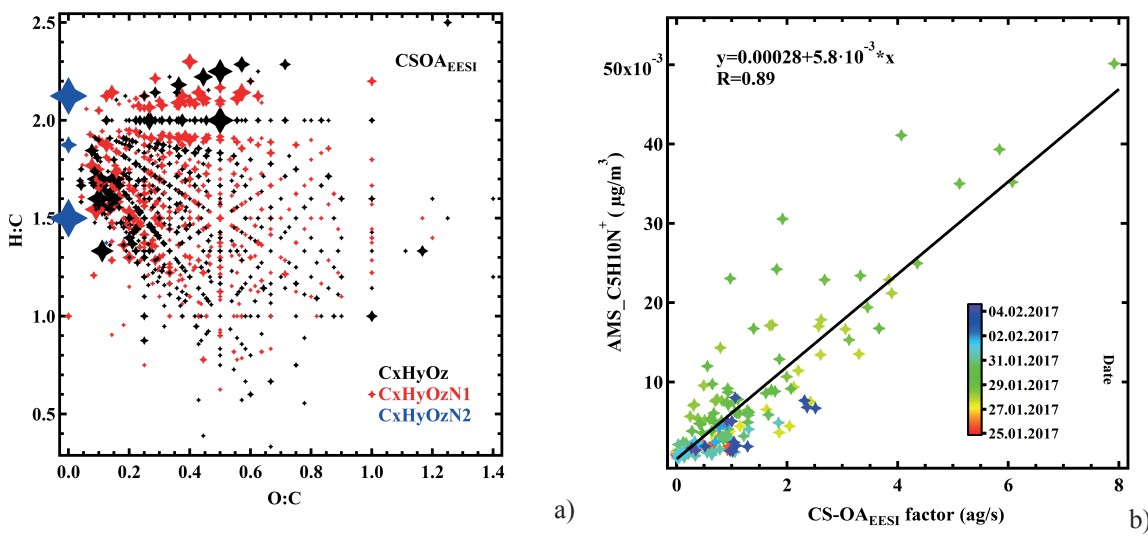

5   **Fig. 5. a) Van Krevelen plot (atomic H:C vs. O:C ratio) of the cigarette smoking (CS-OA$_{EESI}$) factor mass spectrum. Points are sized by the fraction of each ion apportioned to CS-OA$_{EESI}$. Colors denote C$x$H$y$O$z$, C$x$H$y$N$_1$O$z$, and C$x$H$y$N$_2$O$z$ groups. b) Comparison of CS-OA$_{EESI}$ and C5H10N$^+_{AMS}$, colored by time.**




**Fig. 6.** Factor profiles (a) and stacked time series (b) of the 6 EESI-TOF SOA PMF factors, together with AMS OOA. The latter panel also shows meteorological data. All EESI-TOF data are plotted in mass flux (ag/s), AMS in µg m$^{-3}$, other units are included. Factor profiles (b) are molecular weighted and are normalized such that the sum of each profile is 1.

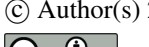



**MABB_LOW**EESI

**MABB_TRANS**EESI

**MABB_HIGH**EESI

**Fig. 7. Wind analysis results using the SWIM model on the concentrations of MABB_LOW$_{EESI}$, MABB_TRANS$_{EESI}$,**
**MABB_HIGH$_{EESI}$. Left: wind direction combined with frequency, wind speed in m/s. Right: the wind speed and wind direction.**





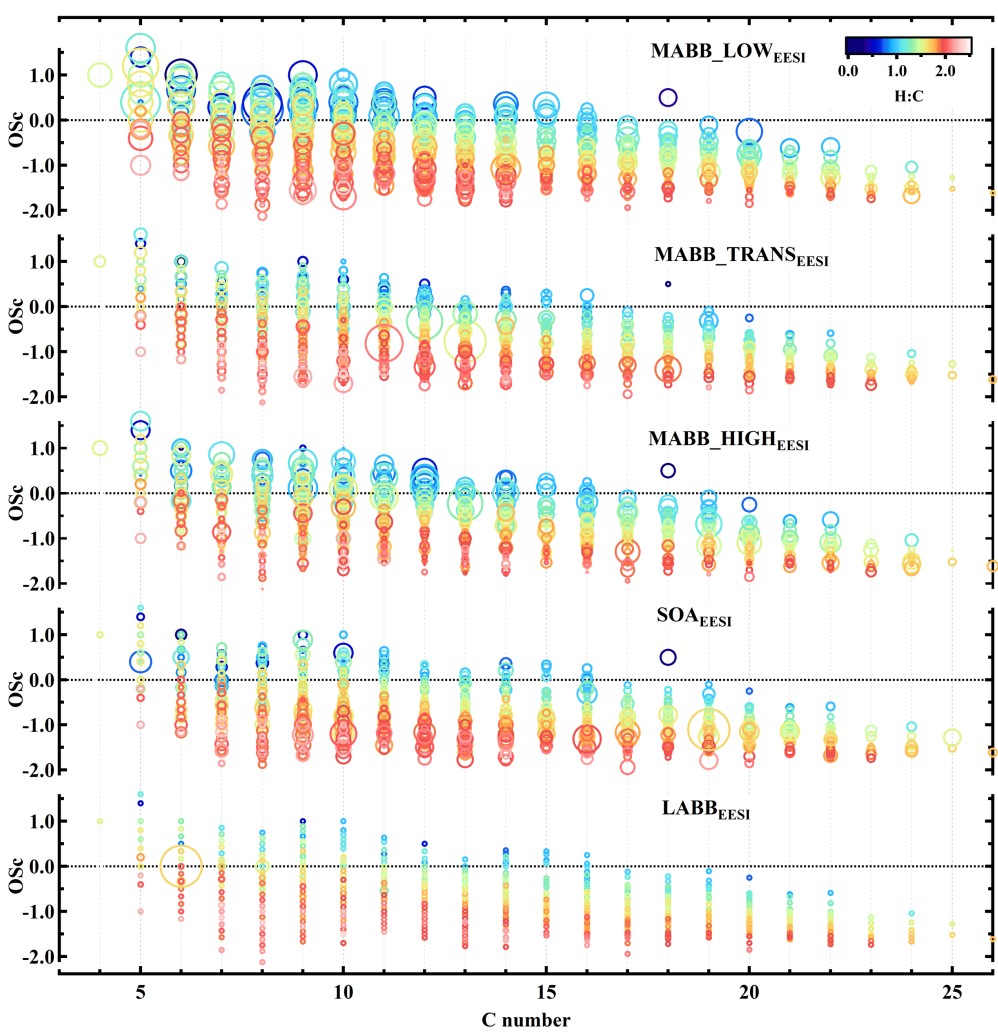

**Fig. 8. Carbon oxidation state (OSc) as a function of number of carbons atoms for the factors, More Aged Biomass Burning_Low temperature, More Aged Biomass Burning_Transition, More Aged Biomass Burning_High temperature, Secondary organic aerosol, Less Aged Biomass Burning. Points are colored by atomic H:C ratio and sized by the fraction of each ion apportioned to the designated factor.**





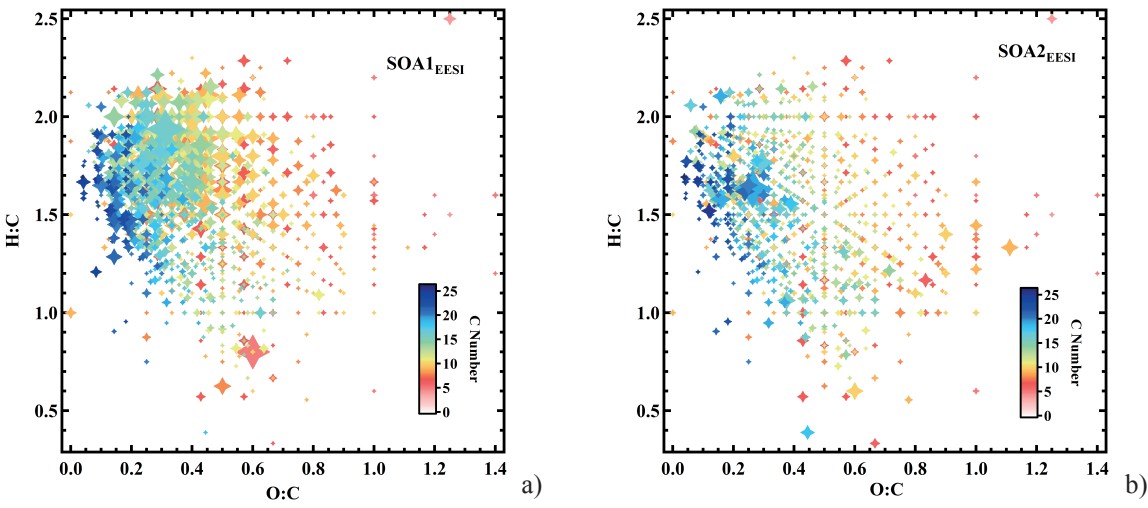

Fig. 9 Van Krevelen plots (atomic H:C vs. O:C) for the SOA1$_{EESI}$ and SOA2$_{EESI}$ factor mass spectra. The points are sized by the fraction of each ion apportioned to SOA1$_{EESI}$ and SOA2$_{EESI}$ and colored by the number of carbon atoms.







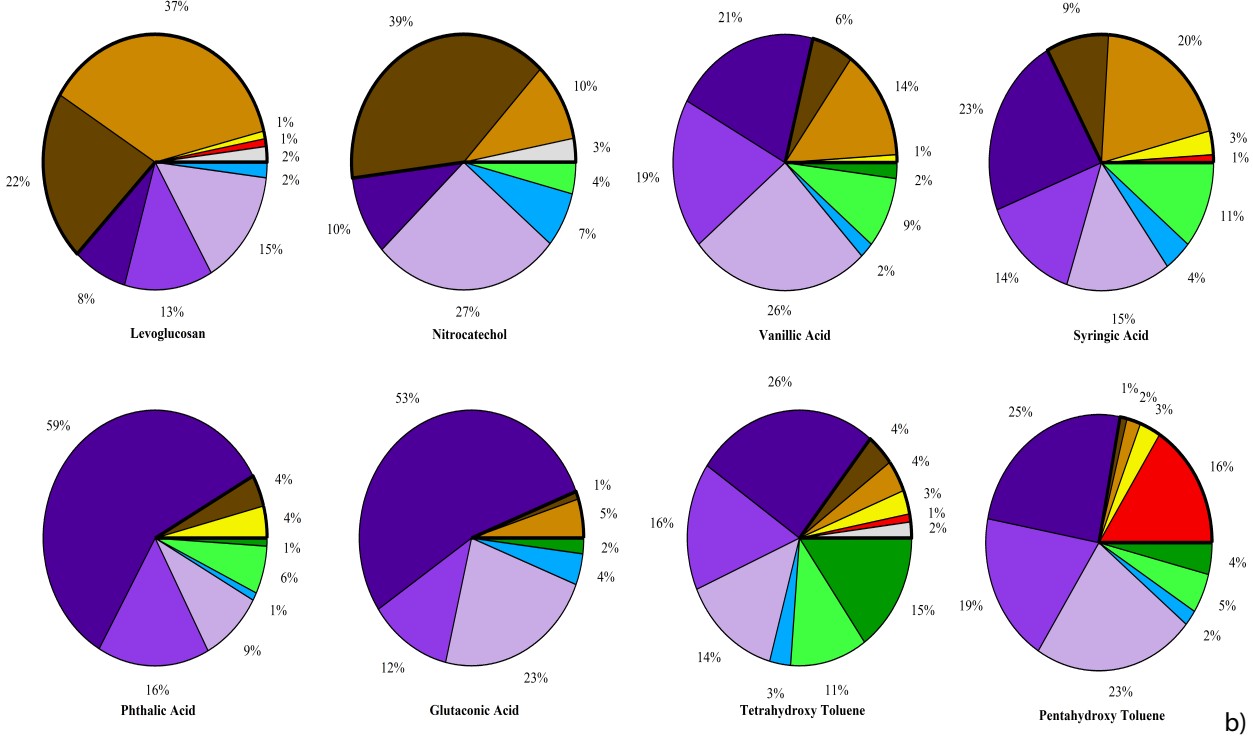

**Fig. 10. Time series of the mass flux (ag/s) of the selected ions found for the EESI-TOF PMF factors (a) and corresponding pie charts (b).**





**Fig. 11. Standardize matrix of individual EESI-TOF ions vs. EESI-TOF PMF factors. Ions and factors are sorted according to the results of their respective hierarchical clustering analysis; the resulting dendrograms are shown on the respective axes. The color of the compounds' groups in the dendrogram are chosen to make groupings convenient to read (color is random).**





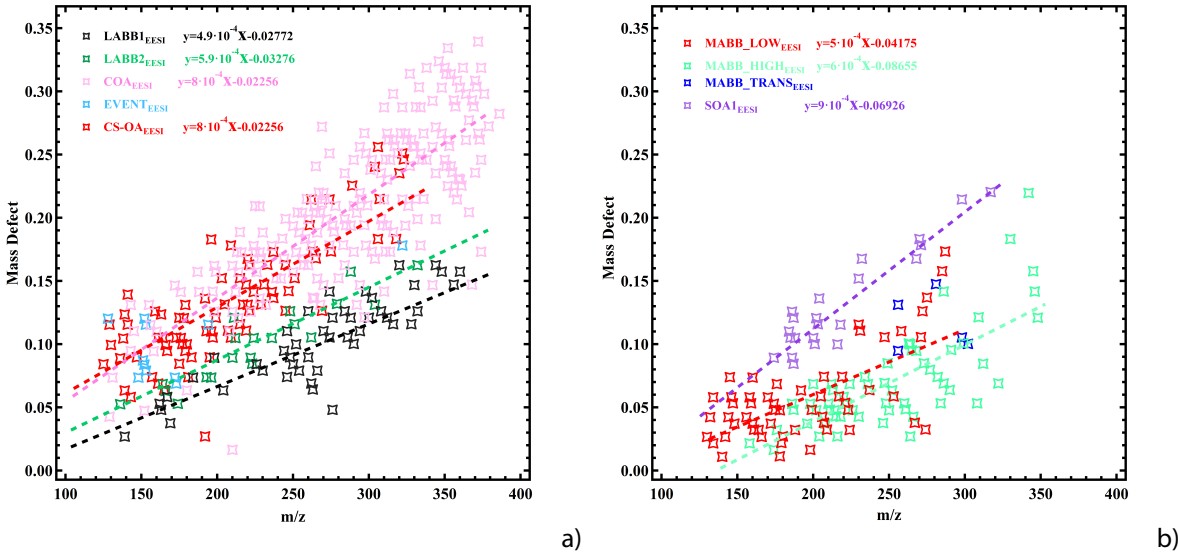

a)

b)

**Fig. 12. Mass defect filtering plot of factor-specific ions (identified from the cluster analysis) for selected EESI-TOF POA (a) and SOA (b) factors.**





**Fig. 13. Comparison of EESI-TOF and AMS. Total EESI-TOF mass flux (ag/s) as a function of AMS OA, points are colored by date (a) and the fraction of levoglucosan (b). The EESI-TOF and AMS comparison in terms of H:C (c) and O:C (d), points are colored by date.**



**Fig. 14. Comparison between EESI factors and AMS factors: time series of the mass flux of each EESI PMF factors (a) and time series of concentrations of each AMS PMF factors (b). Pie charts of source apportionment results from the EESI (left) and AMS (right) (c). The thick block frame denotes the sum of the primary OA for both data sets.**



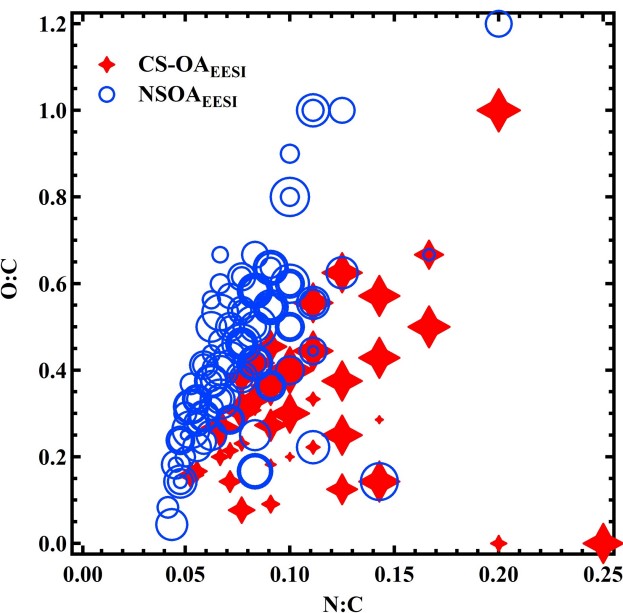

**Fig. 15. The atomic O:C vs. N:C plot of the CS-OA$_{EESI}$ and NSOA$_{EESI}$ factors mass spectra. Points are sized by H:C value of each distinguished ion of the CS-OA$_{EESI}$ and NSOA$_{EESI}$.**

