# Peer review of "Organic aerosol source apportionment in Zurich using an extractive electrospray ionization time-of-flight mass spectrometry (EESI-TOF): Part II, biomass burning influences in winter"

_Atmospheric Chemistry and Physics, 2019_

## Referee Comment (RC1) · Anonymous Referee #1 · 4 Mar 2019

The authors present PMF (ME-2) based source apportionment of organic aerosol measured with AMS and EESI-TOF at an urban background site in Zurich, Switzerland, during a few days in Winter 2017. There exist already quite a few source apportionment studies from this location, which therefore provides a good opportunity to showcase the additional possibilities of the newly developed EESI-TOF, especially related to molecular identification of SOA compounds from different sources. This is an interesting and well-written manuscript and well suited for ACP, and I suggest its publication after addressing the comments below.

[Figure]

General comments

The measurement campaign was very short (January 25 – February 5), so the authors should add a short discussion on the significance and representativeness of their results.

Also a short note should be added on why PMF was done separately for AMS and EESI-TOF data, and if the authors expect results to differ for a combined approach (if possible at all).

Conclusions are a bit meager, and some effort could be taken in better describing the (atmospheric) implications of the results.

A few important references are given as "in prep." (also see specific comments below)– if these references are not available soon the authors should consider removing them and adding more information to the present manuscript.

Specific comments

P. 5, l. 9: What is "most"? Since the paper about the instrument is not available yet, this statement has to be made more quantitative/explicit. Which gas phase species are removed, based on what properties? The denuder only "reduces the gas phase background" – so what is left?

P. 5, l. 12: This might be discussed in the instrument paper, however, as this is not available, a short discussion should be included here: Are artefacts due to extraction to be expected, depending on solvent?

P. 5, l. 15: This implies heating afterwards. Please clarify.

P. 5, l. 26 – 32: Have the authors tried to relate the mass flux to ambient concentrations? Please discuss this. Do the authors expect a simple calibration with levoglucosan to be able to cover "instrument flow rate, EESI extraction/ionization efficiency, declustering probability, and ion transmission"?

P. 6, l. 11 -13: Please include (e.g in the supplementary) more details on error calculations (show data periods chosen, values etc.) This can be very useful for readers / future users.

P. 8, l. 1-2: Why were exactly these factors constrained in the AMS PMF? Please clarify. Does that introduction of subjectivity distort your solution?

P. 11, l. 16 – 18: How do the diel patterns of the nicotine and COA factors compare? Could it be that they are similar due the influence of restaurant opening times, with people gathering outside the restaurants to smoke?

P. 12, l. 6-7: As Figure 2b shows, C8H12O6 has a very prominent signal in the LABB spectra. The authors speculate that this ion represents hydroperoxides from the oxidation of phenolic compounds by OH radicals during daytime. Biomass burning seems to be mostly going on during evening/night times – how come daytime oxidation of compounds primarily emitted at night would have such a bit signal?

P. 12, l. 14-15: Already mention here what this "different" thing is

P. 12, l. 23 – 25: January 27 – 29 was a weekend, and a quick google search revealed that the Zurich game festival (http://www.ludicious.ch/ludicious-2017/) was taking place then, which would mean a lot around Zurich Kaserne, eating, smoking…Also LABB2 is high then, despite higher temperatures. How do the authors explain this?

P. 15, l. 31 – 33: How sure can the authors be that the molecular formulae they measure correspond to the mentioned compounds? Please add a short discussion on this uncertainty.

Technical corrections P. 2, l. 19 – 21: Sentence structure

P. 3, l. 16: Family?

P. 3, l. 23: It would be beneficial if this paper was available once this manuscript is online

P. 4, l. 24: I suggest removing this reference unless the paper is available at the time of publication of this one.

P. 6, l. 1: Number fitting?

P. 6, l. 6: Servo and MS? Specify

P. 7, l. 20: Minimizes

P. 17, l. 7: separately?

Fig. 2, 6: Add arrows/lines to clarify the corresponding stick labels

Fig. 10: The figure looks squished

---

## Referee Comment (RC2) · Anonymous Referee #2 · 12 Mar 2019

General comments:

The manuscript presents an investigation of the performance of a novel technique, extractive electrospray ionization time-of-flight mass spectrometer (EESI-TOF), to measure ambient organic aerosol (OA) composition in real-time and characterize it into a near-molecular level. Factor analysis of molecules identified by the new technique resulted in 11 factors comprising five primary organic aerosol (POA) and six secondary organic aerosol (SOA). EESI-TOF allows a more detailed molecular identification of OA compared to the established aerosol mass spectrometer (AMS), and yet the two

techniques show a good agreement in terms of bulk analysis and groups of factors. The study is well within the scope of Atmospheric Chemistry and Physics journal. The manuscript flows logically, and the figures are clear. I found a few technical issues as listed below. Also, I have some questions and suggestions as described in the following. Finally, I would recommend publishing the manuscript after addressing the comments.

Specific comments:

a) In Figure 2a, time delay seems to exist between LABB1 and LABB2 and levoglucosan ($C_6H_{10}O_5$) for peaks on 28-29 Jan. Also, in Figure 4a, LABB1 has a higher O:C and lower H:C compared to LABB2EESI. These might indicate LABB1 is more oxygenated. It could also originate from a different source from LABB2. How does the wind regression analysis for these factors show?

b) In the discussion of SOA factors (Pg. 15 Lns. 12-15), instead of a time series plot (Figure 6b), a diurnal plot will better show the daytime cycle of SOA2. Also, I am not sure what is the evidence of that both factors (SOA1 and SOA2) are associated with SOA as opposed to the OOA-AMS factors. If it is due to similarities of the SOA factors mass spectra to the monoterpene-related factors (Pg. 15 Lns. 5-6), I think it is more convincing to compare (plot correlation) of the mass spectra.

c) For the nitrogen-containing SOA factor, could the variability in time between the high peak at the night of 3-4 Feb and the small peak at the night of 28-29 Jan associated by a change in temperature, or was it caused by shifting of air masses? What the wind analysis or back-trajectory analysis suggest?

d) The factor dendrogram seems to resolve five groups instead of three (Figure 11); SOA1 and EVENT factors are one group, and SOA2 and NSOA factors are another group. What do these groupings suggest in terms of characteristics? The discussion of factors dendrogram in Pg. 17 Lns. 6-10 could be expanded to include these groups.

[Figure]

Technical comments:

a) Pg. 15 Ln. 21: Ratio of N:C of NSOA in Table S1 is 0.04.

b) Pg. 16 Lns. 13-14: Check the percentage contribution of syringic acid and vanillic acid that are apportioned to MABB factors. Based on Figure 10b, they are supposed to be 52% and 66%, respectively.

c) Pg. 16 Lns. 23-24: Be consistent with the decimal of percentage. The percentage can be off if the decimal is included.

d) Table S2: LAWB1 refers to LABB1? Check the acronyms of factors and make them consistent throughout the main text and supporting information.
* * *

---

## Referee Comment (RC3) · Anonymous Referee #3 · 16 Mar 2019

The paper by Qi et al. is a comprehensive study of wintertime organic aerosol composition and sources, using factorization of mass spectra obtained from two online instruments; one that uses aerosol volatilization + EI (AMS) and one based on extractive electrospray ionization (EESI-MS). This I believe is the first study that reports on field measurements of the EESI-MS and highlights its potential for organic aerosol source apportionment, given its soft ionization and lack of thermal fragmentation. The results are quite interesting, indicating that a large fraction of OA in wintertime in Zurich is derived from fresh and aged biomass burning (this picture is different than what

AMS-based factors suggest since the contribution of BB to AMS-OOA factor cannot be separated from other sources of OOA). The paper is overall well-written (although I still have some editorial suggestions below). There are a lot of figures in the main text (and SI), but given the extent of the presented data, maybe there's no way around it. I recommend publishing the paper after following comments are addressed.

Technical comments: P1, L19: "This suggests the EESI-TOF apportionment can be approximately taken at face value, despite ion-by-ion differences in relative sensitivity." Perhaps this is too strong of a statement since this environment is under major influence of BB (and little HOA) and not a mix of very different sources as can be common in other environments, especially in the summer.

P6, L3: why were the ions smaller than 135 amu small? Was tuning changed on purpose to increase transmission of larger ions? Or is there something different in the mass spec design compared to that of a typical CIMS/AMS?

P6, L8: Just out of curiosity, how stable was the background?

P8, L2: why were different factors constrained for AMS and EESI-ToF?

P10, L1-2: If OOA1 has some of the CxHy fragments, shouldn't that be the more volatile OOA factor?

P15, L3: It's surprising that the SOAEESI factor is less oxidized than the oxidized BB factors. At least in the AMS-based PMF factors, the SOA (OOA) factors are quite more oxidized than the BBOA factor. Do the EESI data suggest that AMS OOA factors may contain the aged BB emissions too? Could the authors elaborate on this. After reading the paper, I realize the authors address this in Section 3.5. It will be good to include a sentence in P15 indicating that this surprising result is going to be further examined.

P17, L27-28: If the slopes of the mass defect vs. m/z points for both less and more aged BB is similar, how can one trust interpreting the slopes to understand types of functional groups added to the molecules included in the other factors?

[Figure]

Section 3.5: I believe this section should be discussed earlier. As I was reading the paper, I kept thinking what if differences in sensitivity of the EESI to different molecules is playing a role in determining the identified factors, so it is reassuring if one reads this section before getting deep into the EESI-based PMF factors. Perhaps the first paragraph of this section that's not discussing the PMF factors yet can be moved to earlier parts of the paper.

P18, L7-9: What's the explanation for a lower sensitivity of EESI for the times levoglucosan was low? Based on the AMS factors, the initial period is not dominated by HOA that EESI is blind to.

Editorial comments: Some correlation coefficients were presented as R, some as R2. Please use one consistently. Quality of some figures was not good when viewed at 100% and the legends/axis labels were blurry.

P 1. L15: "...was derived..."

P1, L21: consider changing "..utilize a...." to "...utilization of a ..."

P2, L14: delete "of" in "...fragments typical SOA molecules".

P6, L2: change "The total number fitting of 1125 ions...." to "The total number of 1125 fitted ions ...".

P6, L9: delete "the" to read " ...were removed from further analysis."

P6, L19: "An HR-ToF-AMS..."

P6, L21: "spent"

P14, L4/5: it should be NOAAMS (not NSOAAMS). Related to this, the discussion on NOAAMS and the EESI-based factor is provided after this sentence, not before.

Fig. 6b- unit for precipitation is not included.

P14, L12: "period 1" should be move to "....study, period 1, occurs ...."

P15, L25: I believe you mean EESI-CHON fragments correlate better NSOA_EESI. I don't think AMS_CHON fragments are displayed.

Fig. 12b. Green color for more aged secondary BB factor is too hard to see against the white background.

P18, L10: consider replacing "big events" to some other phrase

Fig. 13b: intercept of the black line appears to have two negative signs in front

P18, L31: change "primarily" to "primary"

P18, L34: replace "WB-related" by "BB-related"?
* * *

---

## Author Comment (AC1) · 28 May 2019

**Response to the comments of anonymous referee #1**

We thank the referee for the valuable comments which have greatly helped us improve the manuscript. Please find below our responses (in black) after the referee comments (in blue). The changes in the revised manuscript are written in *italic*.

General comments

The measurement campaign was very short (January 25 – February 5), so the authors should add a short discussion on the significance and representativeness of their results.

We agree with the comment that the period of the measurement campaign is short. However, the Zurich-Kaserne site has been extensively characterized in previous studies using AMS and ACSM. The general similarity of the AMS results obtained in the current study to previous publications gives us high confidence that the results obtained here are typical of wintertime conditions at the site (with the exception of the special events clearly separated by the $EVENT_{EESI}$ and $EVENT_{AMS}$ factors).

We add a short note in section 2.1 (P4, L27-31), *"Although the measurement period is relatively short (12 days), the similarity of the AMS results obtained in the current study to previous AMS and ACSM measurements at the same site (Lanz et al., 2007, Canonaco et al., 2013, Richard et al., 2011, Daellenbach et al., 2016) give us high confidence that the sampled aerosol is representative of typical wintertime conditions. Exceptions to this are resolved by the source apportionment into unique event-driven factors, as discussed in the results section."*

Also a short note should be added on why PMF was done separately for AMS and EESI-TOF data, and if the authors expect results to differ for a combined approach (if possible at all).

A combined AMS/EESI-TOF PMF analysis is potentially of high interest and may facilitate quantitative interpretation of EESI-TOF data. However, such combined analyses are highly complex, requiring careful balancing of the explained variability within the two component datasets (Slowik et al., 2010; Crippa et al., 2013). In addition, such combined analyses tend to decrease the ability of the component datasets to retrieve factors resolvable by only a single instrument (such as $HOA_{AMS}$, or factors driven by chemical signatures observable only by the EESI-TOF). As a result, we focus here on exploring the ability of the novel EESI-TOF measurements to improve factor separation and design the source apportionment analysis to maximize this potential.

As requested, we address this in the manuscript (section 2.3) (P8, L8-9) as follows:

*"Execution of PMF analysis on separated AMS and EESI-TOF datasets minimizes the complexity of the analysis, while maximizing the factor resolution ability of the EESI-TOF."*

Conclusions are a bit meager, and some effort could be taken in better describing the (atmospheric) implications of the results.

The following text has been added to the conclusion (P21, L1-4), *"Comparisons of bulk measurements, as well as of individual factors or groups of factors between the EESI-TOF and AMS indicate good agreement, but with the differences in elemental ratios. This suggests that, despite significant uncertainties in the relative response*

*factors of individual ions measured by the EESI-TOF, responses at the level of the PMF factors are relatively similar, with the main differences resulting from the high sensitivity to levoglucosan in the EESI. Furthermore, source apportionment of EESI-TOF provides more classification of SOA factors, separating EESI biomass burning factors as more/ less aged instead of primary / secondary, identifying organic nitrogen containing factors as primary-dominated nitrogen factor / organonitrate-containing secondary factor, which are not possible for AMS PMF."*

A few important references are given as "in prep." (also see specific comments below) – if these references are not available soon the authors should consider removing them and adding more information to the present manuscript.

We agree with the comment. In this manuscript, we presented four "in prep" references, and we modified as bellow:

1. "Lopez-Hilfiker et al., in prep", now the paper is public on AMTD, so it is cited as "==Lopez-Hilfiker et al., 2019==".
2. "Stefenelli et al. in prep", the paper is now online in ACPD, so it is cited as "==Stefenelli et al., 2019==".
3. "Bertrand et al., in prep", the paper has been submitted, so here we change it to "==submitted==".
4. "Lu et al., in prep", status is unchanged.

Specific comments

P. 5, l. 9: What is "most"? Since the paper about the instrument is not available yet, this statement has to be made more quantitative/explicit. Which gas phase species are removed, based on what properties? The denuder only "reduces the gas phase background" – so what is left?

The referenced paper is now available from Atmospheric Measurement Techniques (http://doi.org/10.5194/amt-2019-4), and ==the reference has been updated.== As a result, we have not otherwise modified the current paper but summarize here for reference.

The denuder has not been fully characterized on a compound-by-compound basis, but removes most gas-phase organics with high efficiency (e.g. pinonic acid, 99.6%).

Otherwise, the main source of background (i.e. non-particle-derived) signal is the working solution. This includes a variety of ions related to the NaI dopant and its clusters with acetonitrile and/or water. However, impurities in the working solution also generate detectable background signal. Finally, particles can pass through the denuder but then deposit on a surface rather than be extracted in the spray. Semi-volatile material desorbing from such deposited particles constitutes an additional source of background.

P. 5, l. 12: This might be discussed in the instrument paper, however, as this is not available, a short discussion should be included here: Are artefacts due to extraction to be expected, depending on solvent?

As noted in the previous comment, the instrument paper is now available and this issue is discussed in detail there. Briefly, the principal artifacts deriving from the use of the water, acetonitrile solvent are the potential for clusters of analyte ions with acetonitrile. These are weakly-bound clusters, and their prevalence depends strongly on voltage settings (i.e. collision energy) in the ion transfer optics. Formation of these clusters was found to be negligible at the settings used in the current study. We have clarified this as follows (P5, L25-26):

*"Depending on voltage settings in the ion transfer optics (i.e. collision energy), clusters with acetonitrile can potentially be detected, however these clusters were observed to be negligible during the current study."*

P. 5, l. 15: This implies heating afterwards. Please clarify.

This is discussed in detail in the instrument paper and clarified in the manuscript as follows (P5, L20-22):
*"The droplets then enter the mass spectrometer through a capillary heated to 250 C, however, the very short residence time in this capillary means that the effective temperature experienced by the analyte is much lower and no thermal decomposition is observed."*

P. 5, l. 26 – 32: Have the authors tried to relate the mass flux to ambient concentrations? Please discuss this. Do the authors expect a simple calibration with levoglucosan to be able to cover "instrument flow rate, EESI extraction/ionization efficiency, declustering probability, and ion transmission"?

The reviewer raises two issues here: (1) assessment of the EESI-TOF mass flux in terms of reference measurements, and (2) utility of the levoglucosan calibration. These points are discussed separately below.

The comparison of EESI-TOF mass flux and ambient OA concentrations is the subject of section 3.5 and Fig. 13. Fig. 13a presents the correlation 0.94 between the EESI-TOF mass flux and ambient concentrations. Figures 13c and 13d respectively show the O:C and H:C atomic ratios for the EESI-TOF as a function of those for the AMS. The EESI-TOF and AMS O:C ratios are correlated (R=0.62), however, the O:C ratios estimated by the EESI-TOF are systematically higher than those measured by the AMS. For H:C ratios, we do not observe a correlation. The EESI-TOF values are scattered around approximately 1.56, independent of the AMS H:C ratios which vary between 1.11 and 1.44. The cause for this discrepancy is not yet understood but may be related to differences in ion relative sensitivity.

As this is already a major section of the manuscript, we assume that the reviewer's question was triggered by the sequence of discussion in the original manuscript rather than a general inadequacy in section 3.5. As a result, we have added the following statement to the initial discussion of EESI-TOF mass flux identified by the reviewer:
*"A comparison of the EESI-TOF mass flux to the AMS signal in terms of total signal or mass, bulk properties, and source apportionment results in section 3.5."*

Regarding the second point, here it is important to distinguish between factors affecting the EESI-TOF sensitivity that are ion-dependent, and those which act uniformly across all ions. We use the levoglucosan calibration only to assess the stability of the instrument with respect to the second category (e.g. flow rate, effects of geometric overlap between aerosol and spray droplets on extraction efficiency, effective primary ion concentrations). Ion-specific considerations (extraction/ionization efficiency and ion transmission) cannot be characterized through this simple calibration. However, these are expected to be fundamental properties of the detected ions (in combination with specific instrument settings which are unchanged throughout the study (e.g. voltages in the ion optics), and although unknown are thus assumed to remain constant.

The statement on levoglucosan calibration has been revised for clarity as follows (P6, L7-8):

*"EESI-TOF stability and linearity with mass were confirmed by periodic measurement of nebulized levoglucosan aerosol with quantification of the mass concentration with an SMPS."*

P. 6, l. 11 -13: Please include (e.g in the supplementary) more details on error calculations (show data periods chosen, values etc.) This can be very useful for readers / future users.

The original error matrix includes mass spectra from both direct ambient sampling and filter blank processed with Tofware. Then filer periods were interpolated to yield an estimated background spectrum during ambient measurements. We describe and add more clear detail in the main text as following (P6, L19-24):

*"The corresponding error matrix $\sigma_{ij}$, which has the same dimensions as the data matrix, follows the model of Allan et al. (2003), which calculation includes the uncertainty deriving from electronic noise, ion-to-ion variability at the detector and ion counting statistics. The error estimates in this case incorporate the uncertainties related to both the ambient measurements ($\delta_i$)* (direct ambient sampling period) *and the background ($\beta_{ij}$)* (filter blank measuring period, both are processed with Tofware)*, which are combined in quadrature according to Eq. 2:"*.

P. 8, l. 1-2: Why were exactly these factors constrained in the AMS PMF? Please clarify. Does that introduction of subjectivity distort your solution?

First, we correct a small mistake in the original manuscript, where it was stated that both traffic and cooking factors were constrained in the AMS PMF analysis whereas in fact only traffic was constrained.

With regards to the proposed introduction of subjectivity, we note that it is well-established that factor constraints select specific solutions (i.e., the selection environmentally reasonable subset) from a large set of solutions of approximately equal mathematical quality. These solutions may not be operationally accessible during analysis in the absence of factor constraints (or other rotational control allowing multidimensional exploration; note that rotation via the global fpeak parameter is insufficient). As a result, the solution returned by unconstrained PMF analysis is itself subject to distortion, as its selection by the model from among other solutions of similar quality is effectively arbitrary. Factor constraints address this problem and have been shown to significantly improve PMF model performance by minimizing such arbitrary distortions (Canonaco et al., 2013, Elser et al., 2016).

In the current study, the a-value for HOA was selected according to the correlations between the time series of HOA with the traffic species NOx (P10, L9-10).

P. 11, l. 16 – 18: How do the diel patterns of the nicotine and COA factors compare? Could it be that they are similar due the influence of restaurant opening times, with people gathering outside the restaurants to smoke?

This is a good point and likely contributes to the necessity for constraining the CS-OA$_{EESI}$ to obtain a clear separation. We have added diurnal plots of the EESI-TOF factors to the supplement (Fig. S6). The following statement has been added to the manuscript (P12, L9-13):

*"The difficulty in separating these factors, despite their expected chemical differences, is likely due to strong temporal correlation between cooking and cigarette-smoking emissions due to the proximity of local restaurants (Fig. S6, the diurnal patterns of nicotine and COA$_{EESI}$ factors), where people gather outside to smoke during mealtimes. We therefore attempted to obtain a clean cigarette smoking signature from the dataset to serve as an anchor profile with which to constrain this source."*

P. 12, l. 6-7: As Figure 2b shows, C8H12O6 has a very prominent signal in the LABB spectra. The authors speculate that this ion represents hydroperoxides from the oxidation of phenolic compounds by OH radicals during daytime. Biomass burning seems to be mostly going on during evening/night times – how come daytime oxidation of compounds primarily emitted at night would have such a bit signal?

Here, we made a mistake. This was an early interpretation, but forgot to revise in the text. Oxidation typically leads to a large set chemically related compounds because all these reactive pathways branch in complex ways. In contrast, strong isolated peaks (e.g. levoglucosan) are more likely to result from a specific emissions source and/or process (e.g., because cellulose is a polymer, its pyrolysis leads to a relatively small number of discrete major products including levoglucosan). Although we are unsure of the compound(s) comprising C8H12O6, it is very likely to be primary and not an oxidation product, since it is observed as an isolated peak with high relative intensity.
The incorrect interpretation is deleted in the text.

P. 12, l. 14-15: Already mention here what this "different" thing is

This statement related to the lack of correlation observed between primary/less aged AMS vs. EESI-TOF wood burning factors at the start of campaign, and has been clarified as follows (P13, L8-11):

*"Fig. S8 compares the BBOA$_{AMS}$ factor (Fig. 2a) with LABB1$_{EESI}$, LABB2$_{EESI}$, and the sum of LABB1$_{EESI}$ + LABB2$_{EESI}$, with R 0.59, 0.79, and 0.82, respectively. The correlation is generally good except during the first part of the campaign (25 January to 27 January) which as discussed later relates to the complexity of wood burning classification between the EESI-TOF and AMS."*

P. 12, l. 23 – 25: January 27 – 29 was a weekend, and a quick google search revealed that the Zurich game festival (http://www.ludicious.ch/ludicious-2017/) was taking place then, which would mean a lot around Zurich Kaserne, eating, smoking: : :Also LABB2 is high then, despite higher temperatures. How do the authors explain this?

Thanks for the significant information. The following statement is added to the manuscript (P14, L3-5):
*"The Zurich game festival was taking place at the weekend (the event is apparently held in a building on the SW side of the courtyard), though no human activities in the immediate vicinity of the sampling inlet were evident by inspection of the on-site camera."*

LABB2 is the more event-driven WB and as such is likely not connected to regular (approximately temperature-driven) domestic heating but rather the activities of the large number of people nearby participating in this event. Probably some local wood burning is associated with this.

P. 15, l. 31 – 33: How sure can the authors be that the molecular formulae they measure correspond to the mentioned compounds? Please add a short discussion on this uncertainty.

We agree that this section requires clarification. In particular, the implication that the EESI-TOF identifies specific molecules is misleading, as the instrument can provide only a molecular formula. In some cases, e.g. $C_6H_{10}O_5$, we know that at a minimum several chemically similar isomers are present (i.e., not only levoglucosan but other sugars such as mannosan and galactosan). We have modified both the labels and caption of Fig. 10, as well as the accompanying text, to clarify this point.

Figure labels now highlight the molecular formula, for example: "$C_6H_{10}O_5$" (~ levoglucosan).

The revised Fig. 10 caption is as follows:
*"Fig. 10. Apportionment of selected ions by EESI-TOF PMF. (a) Time series of the mass flux (ag s$^{-1}$) and (b) mean fraction apportioned to each factor. Each ion is associated with a compound of interest having this molecular formula, however, the relative isomeric abundance of this compound cannot be confirmed by the EESI-TOF."*

Revised discussion (P16, L31):
*"Here we investigate the apportionment of eight ions associated with compounds of interest: $C_6H_{10}O_5$ (approximately assigned to levoglucosan), $C_7H_7NO_4$ (methyl-nitrocatechol), $C_9H_{10}O_5$ (syringic acid), $C_8H_8O_4$ (vanillic acid), $C_8H_6O_4$ (phthalic acid), $C_5H_6O_4$ (glutaconic acid$C_7H_8O_4$ (tetrahydroxy toluene) and $C_7H_{10}O_5$ (pentahydroxy toluene). Note that because the EESI-TOF can provide only a molecular formula, we cannot establish for certain the identity of a compound or assess the relative isomeric abundances. For example, $C_6H_{10}O_5$ is likely to consist not only of levoglucosan, but also other sugars such as mannosan and galactosan. The named compounds are thus provided for reference, but their identification should not be considered as conclusive and the ions cannot be assumed to be isomerically pure. Nevertheless, as these assignments are based on molecular*

*investigations of wood burning-related emissions they are likely to be qualitatively correct and provide a useful framework for interpreting molecular aspects of the source apportionment results."*

Technical corrections

P. 2, l. 19 – 21: Sentence structure

This sentence was modified in response to a comment by Reviewer #3, and now reads as follows (P2, L18-20):

*"This suggests the EESI-TOF apportionment in the current study can be approximately taken at face value, despite ion-by-ion differences in relative sensitivity."*

P. 3, l. 16: Family?

Done. We change to GC-family (P3, L16).

P. 3, l. 23: It would be beneficial if this paper was available once this manuscript is online

Done. The instrument paper "An Extractive Electrospray Ionization Time-of-Flight Mass Spectrometer (EESI-TOF) for online measurement of atmospheric aerosol particles" now is public on Atmospheric Measurement Techniques Discussions (AMTD, http://doi.org/10.5194/amt-2019-45).

P. 4, l. 24: I suggest removing this reference unless the paper is available at the time of publication of this one.

We believe there is small typo here, and the reviewer refers to a reference at P.4. l. 14. This paper is now available at ACPD, and the reference has been updated.

P. 6, l. 1: Number fitting?

Done (P6, L9-10). *"The total number of 1125 fitted ions* *(including 882 Na$^+$ adducts, one H$^+$ adduct, and 242 unknown ions) between m/z 135 and 400 were identified."*

P. 6, l. 6: Servo and MS? Specify

The revised text reads (P6, L11-14):
*"Data were pre-averaged to 1 min time resolution, and high resolution peak fitting was performed. Individual 1-min spectra were classified as either ambient measurements, background sampling (through the particle filter), or transitional measurements immediately after switching between ambient/background sampling. Transitional measurements were excluded from further analysis."*.

P. 7, l. 20: Minimizes

Done (P7, L29). *"Equation (3) is solved using a least squares algorithm that iteratively ==minimizes== the quantity Q (Eq. 4),..."*

P. 17, l. 7: separately?

We have revised the text for clarity (P18, L12-19):

*The factor dendrogram identifies several groups of EESI-TOF PMF factors consistent with the interpretations provided above: (1) more aged biomass burning factors (MABB_LOWEESI, MABB_TRANSEESI and MABB_HIGHEESI), (2) less aged biomass burning factors (LABB1EESI and LABB2EESI), and (3) the cooking-related OA and cigarette smoking OA factors. The more aged and less aged biomass burning factor groups are themselves likewise grouped. This clustering is consistent with our interpretation of these factors, as discussed in the previous section. Ions are clustered to different groups using the standardized values. In each factor, there are distinguished molecules (lists of the specific ions (standardized value above 1.5) for each factor is shown in Table S2). The other two resolved groups, one group including SOA1 and EVENT factor, one group containing SOA2 and NSOA factor, apparently don't retrieve the common ions, which make less sense for the current study."*

Fig. 2, 6: Add arrows/lines to clarify the corresponding stick labels

Done. The arrows are added to the figures.

Fig. 10: The figure looks squished

Done. The shape of the figure is changed.

---

## Author Comment (AC2) · 28 May 2019

**Response to the comments of anonymous referee #3**

We thank the referee for the valuable comments which have greatly to helped us improve the manuscript. Please find below our responses (in black) after the referee comments (in blue). The changes in the revised manuscript are written in *italic*.

Technical comments:
P1, L19: "This suggests the EESI-TOF apportionment can be approximately taken at face value, despite ion-by-ion differences in relative sensitivity." Perhaps this is too strong of a statement since this environment is under major influence of BB (and little HOA) and not a mix of very different sources as can be common in other environments, especially in the summer.

We have revised this statement to note that it applies specifically to the range of aerosol composition observed in the current study. We agree that similar analyses are required in different environments before a general conclusion can be drawn. However, even in the current study there is significant variation in the chemical composition (e.g., primary vs. secondary biomass burning, as well as SOA factors with terpene-like signatures that comprise nearly 20 % of the EESI-TOF signal).

The revised statement reads as follows (P2, L18-20):
*"This suggests that the EESI-TOF source apportionment in the current study can be approximately taken at face value, despite ion-by-ion differences in relative sensitivity."*

P6, L3: why were the ions smaller than 135 amu small? Was tuning changed on purpose to increase transmission of larger ions? Or is there something different in the mass spec design compared to that of a typical CIMS/AMS?

For EESI-TOF, we aim to focus on the molecular compositions. The quads were operated such that the transmission decreases rapidly below approximately m/z 150.

P6, L8: Just out of curiosity, how stable was the background?

In this campaign, the background measurement is in the same level (almost the same value) during the whole measurement period.

P8, L2: why were different factors constrained for AMS and EESI-ToF?

Factors may be constrained to overcome a variety of issues compromising an unconstrained PMF solution, including factors with strong temporal correlation, two or more factors with similar chemical signatures, or factors with chemical features that due to the overall chemical variability within the dataset are not clearly mathematically

resolved. In the first case (temporal correlation), the reviewer is correct that one would expect similar factors to be constrained between AMS and EESI-TOF (with the exception of $HOA_{AMS}$, which is primarily composed of hydrocarbon species undetectable by the EESI-TOF). However, the remaining two cases depend on the nature of the chemical measurement by the AMS and EESI-TOF, which are fundamentally different. Therefore, the constrained factors are expected to vary between instruments. For example, in the current analysis, we note that the extensive fragmentation in the AMS results in both the long-chain hydrocarbons observed $HOA_{AMS}$ and the fatty acids in $COA_{AMS}$ yielding $C_xH_y^+$ fragments, which makes these factors appear chemically similar (and requiring constraints). In contrast, the EESI-TOF directly detects cooking-derived fatty acids, making this factor appear chemically unique (and not requiring constraint). Meanwhile, the increased chemical specificity of the EESI-TOF allows identification of a cigarette-smoke related factor based largely on the distinctive $C_{10}H_{14}N_2H^+$ ion, with the separation optimized by factor constraints, whereas the reduced chemical specificity necessitated the incorporation of cigarette smoke into a mixed factor.

We clarify this issue in the manuscript as follows (P8, L10-14):
 *"Different factors were constrained in the two datasets due to the fundamental differences between the AMS and EESI-TOF measurements. Specifically, the absence of fragmentation in the EESI-TOF allowed clear separation of cooking without the need for constraints, while separation of a cigarette smoke factor was only achieved for the EESI-TOF. In addition, constraining an AMS cigarette smoke factor was attempted but failed."*

P10, L1-2: If OOA1 has some of the CxHy fragments, shouldn't that be the more volatile OOA factor?

We agree with this comment.
The OOA1 factor is the less oxygenated factor (LO-OOA), while the current OOA2 factor is more oxygenated OOA (MO-OOA). We had fixed all the corresponding text and figures.

P15, L3: It's surprising that the SOAEESI factor is less oxidized than the oxidized BB factors. At least in the AMS-based PMF factors, the SOA (OOA) factors are quite more oxidized than the BBOA factor. Do the EESI data suggest that AMS OOA factors may contain the aged BB emissions too? Could the authors elaborate on this. After reading the paper, I realize the authors address this in Section 3.5. It will be good to include a sentence in P15 indicating that this surprising result is going to be further examined.

We agree with this suggestion.
Here we add a short sentence to guide readers (P16, L14-15), *"The more detailed comparison between EESI-TOF_SOA factors and AMS_OOA factors will be discussed in Sect. 3.5."*

P17, L27-28: If the slopes of the mass defect vs. m/z points for both less and more aged BB is similar, how can one trust interpreting the slopes to understand types of functional groups added to the molecules included in the other factors?

We have clarified this discussion. The main point in these mass defect plots is that the BB-related factors (both LABB and MABB) exhibit slopes (and intercepts) that are significantly lower than those of the other investigated factors, suggesting increased aromaticity. Within the BB-related factors, LABB2$_{EESI}$ and MABB_HIGH$_{EESI}$ have similar slopes, which are slightly higher than LABB1$_{EESI}$ and MABB_LOW$_{EESI}$.

*Section 3.4 was revised as follows (P18-19):*

*"LABB1$_{EESI}$ and LABB2$_{EESI}$ have a lower mass defect and shallower slope than COA$_{EESI}$ and CS-OA$_{EESI}$, consistent with increased aromaticity. The slopes are $(4.9\pm0.4)\cdot10^{-4}$, $(5.9\pm0.6)\cdot10^{-4}$, $(8\pm0.5)\cdot10^{-4}$ and $(8\pm0.3)\cdot10^{-4}$ for LABB1$_{EESI}$, LABB2$_{EESI}$, COA$_{EESI}$ and CS-OA$_{EESI}$, respectively. The slopes of the two LABB factors as well as those of COA$_{EESI}$ and CS-OA$_{EESI}$ are very similar to each other and have a high possibility to be consistent with CH addition for the former (i.e. $C_{10+x}H_{14+x}O_{4-5}$, theoretical slope $6\cdot10^{-4}$), and CH2 addition for the latter (i.e. $C_{10+x}H_{20+2x}O_{3-5}$ for COA$_{EESI}$ and $C_{10+y}H_{15+2y}NO_{3-5}$ for CS-OA$_{EESI}$ as nearly every CS-OA-specific ion contains a single N atom, theoretical slope $1.1\cdot10^{-3}$).*

*The MABB and LABB factors have similar slopes, despite different ion lists. The slopes of two MABB factors $(0.9\cdot10^{-3})$, as shown in Fig. 12b, are consistent with the addition of CHO functionality (theoretical slope $= 0.1\cdot10^{-2}$). Due to the high variability of the slopes of the MABB factors, it may also contain the other potential possibility for the added functionalities."*

Section 3.5: I believe this section should be discussed earlier. As I was reading the paper, I kept thinking what if differences in sensitivity of the EESI to different molecules is playing a role in determining the identified factors, so it is reassuring if one reads this section before getting deep into the EESI-based PMF factors. Perhaps the first paragraph of this section that's not discussing the PMF factors yet can be moved to earlier parts of the paper.

We agree with the reviewer that molecule-dependent sensitivity of the EESI-TOF is an important point and that the reader should be aware from the start that an AMS/EESI-TOF comparison is presented. With that said, we prefer to retain the current order of sections because the bulk AMS/EESI-TOF comparisons cannot be well understood without also considering the factor-by-factor comparisons, and the factor comparisons in turn cannot be understood without first presenting the PMF analysis. However, we have added the following text to the beginning of section 3 to clarify the discussion structure (P9, L21-26):

*"Results of AMS and EESI-TOF PMF analyses are presented in sections 3.1 and 3.2, respectively. Section 3.3 focuses on the EESI-TOF PMF results are then exploited to assess the apportionment of specific ions related to key marker compounds (section 3.3) and to identify groups of molecules uniquely characteristic of the retrieved factors (section 3.4). However, quantitative interpretation of the EESI-TOF PMF results is complicated by differences in the relative sensitivity of the EESI-TOF to different molecules. Therefore section 3.5 presents a comparison of the EESI-TOF and AMS results in terms of total signal, bulk atomic composition, and relative apportionment to the different factors."*

P18, L7-9: What's the explanation for a lower sensitivity of EESI for the times levoglucosan was low? Based on the AMS factors, the initial period is not dominated by HOA that EESI is blind to.

Laboratory measurements indicate that the EESI-TOF is likely more sensitive to levoglucosan than to typical SOA components (Lopez-Hilfiker et al., 2019). This has been clarified in the text as follows (P19, L16-19):

*"An SOA-dominated period with low levoglucosan concentrations (red line) toward the beginning of the campaign exhibits a lower sensitivity than during a period with higher levoglucosan concentrations (black line), which includes the events on 28.01.2017 and 29.01.2017 characteristic of EVENT$_{EESI}$ (Lopez-Hilfiker et al., 2019)."*

Editorial comments: Some correlation coefficients were presented as R, some as R2. Please use one consistently.

Done. We change all the R2 to R. R2 is focus on the section 3.2.2, LABB factors.

Quality of some figures was not good when viewed at 100% and the legends/axis labels were blurry.

Done. Fig. 10 and Fig. 12 are changed to the high resolution.

P 1. L15: ": : :was derived: : :"

Done. *"While the AMS attributed slightly over half the OA mass to SOA but did not identify its source, the EESI-TOF showed that most (> 70 %) of the SOA was derived from biomass burning." (P2, L15)*

P1, L21: consider changing "..utilize a: : :." to ": : :utilization of a : : :"

Done. *"The apportionment of specific ions measured by the EESI-TOF (e.g. levoglucosan, nitrocatechol, and selected organic acids), and utilization of a cluster analysis-based approach to identify key marker ions for the EESI-TOF factors are investigated." (P2, L21)*

P2, L14: delete "of" in ": : :fragments typical SOA molecules".

Done. *"The chemical analysis of aerosol online-proton transfer reaction mass spectrometer (CHARON-PTR-MS) has no significant thermal decomposition but the ionization scheme fragments typical SOA molecules." (P3, L15-16)*

P6, L2: change "The total number fitting of 1125 ions: : :." to "The total number of 1125 fitted ions : : :".

Done. *"The total number of 1125 fitted ions (including 882 Na$^+$ adducts, one H$^+$ adduct, and 242 unknown ions) between m/z 135 and 400 were identified." (P6, L9-10)*

P6, L9: delete "the" to read " : : :were removed from further analysis."

Done. *"Ions with a mean signal-to-noise ratio (SNR) below 2 were removed from further analysis." (P6, L17-18)*

P6, L19: "An HR-ToF-AMS: : :"

Done. *"An HR-TOF-AMS was deployed for online measurements of non-refractory (NR) $PM_{2.5}$." (P6, L28)*

P6, L21: "spent"

Done. *"The AMS recorded data with 1 min time resolution, of which 30 s was spent recording the ensemble mass spectrum (mass spectrum (MS) mode) and 30 s recording size-resolved mass spectra." (P6, L29-31)*

P14, L4/5: it should be NOAAMS (not NSOAAMS). Related to this, the discussion on NOAAMS and the EESI-based factor is provided after this sentence, not before.

Done. In the section 3.1, the AMS source apportionment, we suggest this factor is a mixed factor, so here we don't include it into the calculation of AMS OOA contribution. *"$NOA_{AMS}$ is excluded from this calculation due to the contribution from primary cigarette smoke discussed above." (P15, L2-3)*

Fig. 6b- unit for precipitation is not included.

Done. The unit for precipitation has been added (*mm/h*).

P14, L12: "period 1" should be move to ": : :.study, period 1, occurs : : :."

Done. *"As shown in Fig. 6b, the coldest part of the study, period 1, occurs from 25 to 27 January." (P15, L9-10)*

P15, L25: I believe you mean EESI-CHON fragments correlate better NSOA_EESI. I don't think AMS_CHON fragments are displayed.

Done. *"Figure S10 shows a comparison of the $NSOA_{EESI}$ time series and CS-$OA_{EESI}$ time series with the CHON ions from the EESI and CHN ions from the AMS, respectively. The group of EESI_CHON ions shows the same temporal variation as the $NSOA_{EESI}$ factor (Fig. S10) while the AMS_CHN group is more correlated to the primary organic group." (P16, L24-26)*

Fig. 12b. Green color for more aged secondary BB factor is too hard to see against the white background.

Done. We change to the black color.

P18, L10: consider replacing "big events" to some other phrase

Done. *"An SOA-dominated period with low levoglucosan concentrations (red line) toward the beginning of the campaign exhibits a lower sensitivity than during a period with higher levoglucosan concentrations (black line), which includes the events on 28.01.2017 and 29.01.2017 characteristic of EVENT$_{EESI}$."* (P19, L16-19)

Fig. 13b: intercept of the black line appears to have two negative signs in front

Done. We delete one negative sign.

P18, L31: change "primarily" to "primary"

Done. *"Both AMS and EESI-TOF factors stacked time series (Fig. 14) show clearly that biomass burning is dominated by secondary fractions early in the campaign, mixed fractions in the middle of the campaign, and primary fractions late in the campaign."* (P20, L6)

P18, L34: replace "WB-related" by "BB-related"?

Done. *"…, it is hard to define how much of AMS OOA is BB-related as a function of time."* (P20, L9)

---

## Author Comment (AC3) · 28 May 2019

**Response to the comments of anonymous referee #2**

We thank the referee for the valuable comments which have greatly to helped us improve the manuscript. Please find below our responses (in black) after the referee comments (in blue). The changes in the revised manuscript are written in *italic*.

Specific comments:

a) In Figure 2a, time delay seems to exist between LABB1 and LABB2 and levoglucosan (C6H10O5) for peaks on 28-29 Jan. Also, in Figure 4a, LABB1 has a higher O:C and lower H:C compared to LABB2EESI. These might indicate LABB1 is more oxygenated. It could also originate from a different source from LABB2. How does the wind regression analysis for these factors show?

Concentrations of LABB2 are high only on three days, roughly corresponding to the EVENT$_{EESI}$ factor that we now associate with a local festival (the Zurich game festival, the event is held in a building on the SW side of the courtyard in which the instrument is deployed). As a result, the reviewer's suggestion of a different source is likely correct. This is further supported by wind regression analysis of these two factors, shown below and added to the supplement as Fig. S7. LABB1 does not correspond to a specific wind direction, consistent with local, widespread domestic wood combustion. In contrast, LABB2 originates predominantly from a single wind direction while the smaller source to the SE is on the third day.

The sentences are added to the manuscript (P13, L5-11): *"The wind regression analysis of these two factors are shown in Fig. S7. LABB1$_{EESI}$ does not correspond to a specific wind direction, in contrast, LABB2$_{EESI}$ originates predominantly from a single wind direction, excluding the smaller source to the SE on the third day."*

[Figure]

*Fig. S7. Wind analysis results using the SWIM model on the concentrations of LABB1 (a) and LABB2 (b).*

b) In the discussion of SOA factors (Pg. 15 Lns. 12-15), instead of a time series plot (Figure 6b), a diurnal plot will better show the daytime cycle of SOA2. Also, I am not sure what is the evidence of that both factors (SOA1 and SOA2) are associated with SOA as opposed to the OOA-AMS factors. If it is due to similarities of the SOA

factors mass spectra to the monoterpene-related factors (Pg. 15 Lns. 5-6), I think it is more convincing to compare (plot correlation) of the mass spectra.

This is a good point, we added diurnal plots of the EESI-TOF factors to the supplement (Fig. S6), shown below. We have also added a comparison of the SOAEESI mass spectra with the monoterpene-related SOA factor from the Zurich summer campaign as Fig. S9.

[Figure]

a)

b)

*Fig. S6. The diurnal variation of EESI POA factors (a) and EESI SOA factors (b).*

c) For the nitrogen-containing SOA factor, could the variability in time between the high peak at the night of 3-4 Feb and the small peak at the night of 28-29 Jan associated by a change in temperature, or was it caused by shifting of air masses? What the wind analysis or back-trajectory analysis suggest?

Yes, we agree with the comment, it is possible that the two nitrogen peaks are associated by a change in temperature. The temperature on 3-4 Feb was much higher than 28-29 Jan, but it was enhanced from 1 Feb. We assume the unique time series may also indicate other chemistry or emission process of the nitrogen-containing compounds, which we also plan to further study.

We plot the time series of wind speed and wind direction on 3-4 Feb and 27-29 Jan, shown below. The wind direction during these days are variable, especially on 3-4 Feb, which may not be a clear evidence.

[Figure]

d) The factor dendrogram seems to resolve five groups instead of three (Figure 11); SOA1 and EVENT factors are one group, and SOA2 and NSOA factors are another group. What do these groupings suggest in terms of characteristics? The discussion of factors dendrogram in Pg. 17 Lns. 6-10 could be expanded to include these groups.

We agree with the comment and have clarified the figure description. In the current study, the agglomerative hierarchical clustering is conducted based on the profiles from PMF. The dendrogram is generated with Euclidean distance metric and average linkage, showing relationships between each group and each factor. However, the dendrogram does not directly show which groups are "tight" (i.e. containing closely related factors or elements) and which are loose. With respect to the factors, the original text focused on three of the five groups where the grouping was consistent with factor definitions (implicitly assuming them to be tightly grouped) while not commenting on the other groups (assumed to be more loose).

We add a short illustration in the main text.

The revised text reads (P18, L12-19): *"The factor dendrogram identifies several groups of EESI-TOF PMF factors consistent with the interpretations provided above: (1) more aged biomass burning factors (MABB_LOW$_{EESI}$, MABB_TRANS$_{EESI}$ and MABB_HIGH$_{EESI}$), (2) less aged biomass burning factors (LABB1$_{EESI}$ and LABB2$_{EESI}$), and (3) the cooking-related OA and cigarette smoking OA factors. The more aged and less aged biomass burning factor groups are themselves likewise grouped. This clustering is consistent with our interpretation of these factors, as discussed in the previous section. Ions are clustered to different groups using the standardized values. In each factor, there are distinguished molecules (lists of the specific ions (standardized value above 1.5) for each factor is shown in Table S2). The other two resolved groups, one group including SOA1 and EVENT factor, one group containing SOA2 and NSOA factor, apparently don't retrieve the common ions, which make less sense for the current study."*

Technical comments:

a) Pg. 15 Ln. 21: Ratio of N:C of NSOA in Table S1 is 0.04.

Done (P16, L21). The value in Table S1 (0.04) is correct, and the value in the main text has been fixed.

b) Pg. 16 Lns. 13-14: Check the percentage contribution of syringic acid and vanillic acid that are apportioned to MABB factors. Based on Figure 10b, they are supposed to be 52% and 66%, respectively.

Done (P17, L19). *"..., which in turn are a major component of biomass combustion emissions, and are apportioned primarily to the MABB$_{EESI}$ factors (52 % for syringic acid and 66 % for vanillic acid)."*

c) Pg. 16 Lns. 23-24: Be consistent with the decimal of percentage. The percentage can be off if the decimal is included.

Done (P17, L29). The main text is changed. *"Tetrahydroxy toluene ($C_7H_8O_4$) and pentahydroxy toluene ($C_7H_8O_5$) are apportioned mainly to secondary factors (85 % and 78 %, respectively)."*

d) Table S2: LAWB1 refers to LABB1? Check the acronyms of factors and make them consistent throughout the main text and supporting information.

Done. Yes, it should be *LABB1.* we correct the other names, *LABB2, MABB_LOW, MABB_TRANS, MABB_HIGH, CS-OA.*

---

## Author Response (AR2)

**Response to the comments of co-editor**

We thank the referee for the valuable comments which have greatly helped us improve the manuscript. Please find below our responses (in black) after the referee comments. The changes in the revised manuscript are written in *italic*.

Page 3: I agree with one of the reviewers that papers in preparation should not be cited (to be consistent with the accepted practices). Please remove reference to Lu et al.

Done. We remove the reference. (P3, L33)

Page 16, Line 33: missing parenthesis and comma

Done. We revised it (P16, L33-34). *"Here we investigate the apportionment of eight ions associated with compounds of interest: C6H10O5 (approximately assigned to levoglucosan), C7H7NO4 (methyl-nitrocatechol), C9H10O5 (syringic acid), C8H8O4 (vanillic acid), C8H6O4 (phthalic acid), C5H6O4 (glutaconic acid), C7H8O4 (tetrahydroxy toluene) and C7H10O5 (pentahydroxy toluene)."*

Figure 1: amu is an outdated unit for molecular weight; it is not used for the mass-to-charge ratio in mass spectrometry. The quantity m/z is typically viewed as dimensionless so I would change the X-axis label accordingly, as you did in other mass spectra figures. While you are changing this figure, the yellow font used for the COA factor label is very hard to see, a different color is recommended. Also this figure appears hazy on my screen (the rest of the figures do not have this problem). Perhaps there is a way to export it at a higher resolution.

Done. We changed the label of X-axis to m/z, color of the COA factor, and the resolution of the figures (Fig.1 and Fig. S1).

Figure 3: The figure legend implies that this plot uses H:C ratios for the ions as opposed to neutral species. I think VK plots for the neutrals instead of ions are easier to interpret. Perhaps it is not even an issue given how few H+ adducts (just one?) you observe. Since you are providing formulas of the neutral compounds in the text, why not calculate the H:C for the neutral species? The same comment applies to the rest of the VK plots.

Done. We agree with the comment that the neutrals are easier to interpret and are the whole story. We change the relevant texts and the captions of the figures (Fig. 3, 4, 5, and 9).

Figure 10: the labels is this figure are very small relative to the figure size, I am worried they will be invisible in the final print. IS there any way to increase them? (In general, labels in many of the figures in this paper are disproportionally small, but in this this figure it is extreme).

Done. We replot the figure with increasing the size of the labels.

Figure 13 and Figure 14: I would fix capitalization in the Y-axis title (flux -> Flux)

Done. We revised the Y-axis to "Mass Flux" in Fig. 2, 3, 6, 13, 14 and S6. We noted the inconsistent in terms of ag/s versus ag s$^{-1}$. Then, we use ag s$^{-1}$ everywhere to be consistent with $\mu$g m$^{-3}$. In this case, the Fig. 2, 3, 5, 6, 10, 13, 14, S6, S8, and S10 are revised.

Bozzetti et al. (2016) reference is missing page numbers

Done. We revised it.

Veres et al. (2010) reference is missing page numbers

Done. We revised it.

Zhang et al. (2017b) reference is missing page numbers

Done. We revised it.

The use of DOI numbers is not very consistent – some have Doi in front of them, and some have nothing.

Done. We checked all the reference to make them consistent.

Figure S1, S3, S5: dat -> date

Done. We revised it in Fig. S1, S3 and S5.

Figure S6: Hourly -> Hour of the Day

Done. We revised it in Fig. 3 and Fig. S6.

Table S2: I think the right column title should be "Molecular Weight of the Neutral (Da)". I actually do not see how the calculated molecular weight is useful (because you can get it from the formula) – if you list values you should list the observed m/z values for the Na+ and H+ adducts. This way you could also include ions that you observed but could not assign.

Done. We agree with the comment to use the observed m/z values in the right column and change the title to "Measured m/z". These ions are from the cluster analysis not the whole fitting ions, so the unknown ions are not included.